# MixTailor: Mixed Gradient Aggregation for Robust Learning Against Tailored Attacks

**Ali Ramezani-Kebrya**[*]  *alir@es.aau.dk*
*Aalborg University*

**Iman Tabrizian**[*]  *iman.tabrizian@gmail.com*
*University of Toronto and Vector Institute*

**Fartash Faghri**  *faghri@cs.toronto.edu*
*University of Toronto and Vector Institute*

**Petar Popovski**  *petarp@es.aau.dk*
*Aalborg University*

**Reviewed on OpenReview:** https://openreview.net/forum?id=tqDhrbKJLS

## Abstract

Implementations of SGD on distributed systems create new vulnerabilities, which can be identified and misused by one or more adversarial agents. Recently, it has been shown that well-known Byzantine-resilient gradient aggregation schemes are indeed vulnerable to informed attackers that can tailor the attacks (Fang et al., 2020; Xie et al., 2020b). We introduce MixTailor, a scheme based on randomization of the aggregation strategies that makes it impossible for the attacker to be fully informed. Deterministic schemes can be integrated into MixTailor on the fly without introducing any additional hyperparameters. Randomization decreases the capability of a powerful adversary to tailor its attacks, while the resulting randomized aggregation scheme is still competitive in terms of performance. For both iid and non-iid settings, we establish almost sure convergence guarantees that are both stronger and more general than those available in the literature. Our empirical studies across various datasets, attacks, and settings, validate our hypothesis and show that MixTailor successfully defends when well-known Byzantine-tolerant schemes fail.

## 1 Introduction

As the size of deep learning models and the amount of available datasets grow, distributed systems become essential and ubiquitous commodities for learning human-like tasks and beyond. Meanwhile, new settings such as *federated learning* are deployed to reduce privacy risks, where a deep model is trained on data distributed among multiple clients without exposing that data (McMahan et al., 2017; Kairouz et al., 2021; Li et al., 2020a). Unfortunately, this is only half the story, since in a distributed setting there is an opportunity for malicious agents to launch adversarial activities and disrupt training or inference. In particular, adversarial agents plan to intelligently corrupt training/inference through adversarial examples, backdoor, Byzantine, and tailored attacks (Lamport et al., 1982; Goodfellow et al., 2015; Blanchard et al., 2017; Bagdasaryan et al., 2020; Fang et al., 2020; Xie et al., 2020b). Development of secure and robust learning algorithms, while not compromising their efficiency, is one of the current grand challenges in large-scale and distributed machine learning.

It is known that machine learning models are vulnerable to adversarial examples at test time (Goodfellow et al., 2015). Backdoor or edge-case attacks target input data points, which are either under-represented or

---

[*]Equal contributions.
This work was done while the first three authors were at Vector Institute and the University of Toronto.

unlikely to be observed through training or validation data (Bagdasaryan et al., 2020). Backdoor attacks happen through data poisoning and model poisoning. The model poisoning attack is closely related to Byzantine model, which is well studied by the community of distributed computing (Lamport et al., 1982).

In a distributed system, honest and good workers compute their correct gradients using their own local data and then send them to a server for aggregation. Byzantines are workers that communicate arbitrary messages instead of their correct gradients (Lamport et al., 1982). These workers are either compromised by adversarial agents or simply send incorrect updates due to network/hardware failures, power outages, and other causes. In machine learning, Byzantine-resilience is typically achieved by robust gradient aggregation schemes (Blanchard et al., 2017). These robust schemes are typically resilient against attacks that are designed in advance, which is not a realistic scenario since the attacker will eventually learn the aggregation rule and tailor its attack accordingly. Recently, it has been shown that well-known Byzantine-resilient gradient aggregation schemes are vulnerable to informed and tailored attacks (Fang et al., 2020; Xie et al., 2020b). Fang et al. (2020) and Xie et al. (2020b) proposed efficient and nearly optimal attacks carefully designed to corrupt training (an optimal training-time attack is formally defined in Section 4). A tailored attack is designed with prior knowledge of the robust aggregation rule used by the server, such that the attacker has a *provable* way to corrupt the training process.

Establishing successful defense mechanisms against such tailored attacks is a significant challenge. As a dividend, an aggregation scheme that is immune to tailored attacks automatically provides defense against untargeted or random attacks.

In this paper, we introduce MixTailor, a scheme based on randomization of the aggregation strategies. Randomization is a principal way to prevent the adversary from being omniscient and in this way decrease its capability to launch tailored attacks by creating an ignorance at the side of the attacker. We address scenarios where neither the attack method is known in advance by the aggregator nor the exact aggregation rule used in each iteration is known in advance by the attacker (the attacker can still know the set of aggregation rules in the pool).

The proposed scheme exhibits high resilience to attacks, while retaining efficient learning performance. We emphasize that any deterministic Byzantine-resilient algorithm can be used in the pool of our randomized approach (our scheme is compatible with any deterministic robust aggregation rule), which makes it open for simple extensions by adding new strategies, but the essential protection from randomization remains.

## 1.1  Summary of contributions

- We propose an efficient aggregation scheme, MixTailor, and provide a sufficient condition to ensure its robustness according to a generalized notion of Byzantine-resilience in non-iid settings.[1]

- For both iid and non-iid settings, we establish *almost sure* convergence guarantees that are both stronger and more general than those available in the literature.

- Our extensive empirical studies across various datasets, attacks, and settings, validate our hypothesis and show that MixTailor successfully defends when prior Byzantine-tolerant schemes fail. MixTailor reaches within 2% of the accuracy of an omniscient model on MNIST.

## 1.2  Related work

**Federated learning.** Federated Averaging (FedAvg) and its variants have been extensively studied in the literature mostly as optimization algorithms with a focus on communication efficiency and statistical heterogeneity of users using various techniques such as local updates and fine tuning (McMahan et al., 2017; Li et al., 2020b; Wang et al., 2020b; Fallah et al., 2020). Bonawitz et al. (2017); So et al. (2020) studied secure aggregation protocols to ensure that an average value of multiple parties is computed collectively without revealing the original values. Secure aggregation protocols allow a server to compute aggregated updates without being able to inspect the clients' local models and data. In this work, we focus on tailored training-time attacks and robust aggregation schemes.

---

[1]Non-iid settings refer to settings with heterogeneous data over workers. In this paper, we focus on non-identical and independent settings.

**Data poisoning and model poisoning.** Adversaries corrupt traning via data poisoning and model poisoning. In the data poisoning, compromised workers replace their local datasets with those of their interest (Huang et al., 2011; Biggio et al., 2012; Mei & Zhu, 2015; Alfeld et al., 2016; Koh & Liang, 2017; Mahloujifar et al., 2019; Gu et al., 2019; Bagdasaryan et al., 2020; Xie et al., 2020a; Wang et al., 2020a). Data poisoning can be viewed as a relatively restrictive attack class since the adversary is not allowed to perturb gradient/model updates. In the model poisoning, the attacker is allowed to tweak and send its preferred gradient/model updates to the server (Bhagoji et al., 2019; Bagdasaryan et al., 2020; Wang et al., 2020a; Sun et al., 2021). These model replacement attacks are similar to Byzantine and tailored attacks. In this paper, we focus on tailored training-time attacks, which belong to the class of poisoning availability attacks based on the definition of Demontis et al. (2019). We do not study poisoning integrity attacks (Demontis et al., 2019), and MixTailor is not designed to defend against backdoor or edge-case attacks aiming to modify predictions on a few targeted points.

**Robust aggregation and Byzantine resilience.** Byzantine-resilient mechanisms based on robust aggregations and coding theory have been extensively studied in the existing literature (Su & Vaidya, 2016; Blanchard et al., 2017; Chen et al., 2017; 2018; Yin et al., 2018; Alistarh et al., 2018; El Mhamdi et al., 2018; Damaskinos et al., 2019; Bernstein et al., 2019; Yin et al., 2019; Yang & Bajwa, 2019; Rajput et al., 2019; Baruch et al., 2019; Xie et al., 2019; Pillutla et al., 2022; Li et al., 2019; Xie et al., 2020c; Sohn et al., 2020; Peng & Ling, 2020; Karimireddy et al., 2022; Allen-Zhu et al., 2021; Gorbunov et al., 2022; Zhu et al., 2022).

Under the assumption that the server has access to underlying training dataset or the server can control and arbitrarily transfer training samples across workers, a server can successfully output a correct model (Xie et al., 2019; Chen et al., 2018; Rajput et al., 2019; Xie et al., 2020c; Sohn et al., 2020). However, in a variety of settings including federated learning, such assumptions do not hold. Alternatively, Yin et al. (2018) proposed to use coordinate-wise median (comed). Bernstein et al. (2019) proposed a variant of signSGD as a robust aggregation scheme, where gradients are normalized before averaging, which limits the effect of Byzantines as the number of Byzantines matters rather than the magnitude of their gradients. It is well known that signSGD is not guaranteed to converge (Karimireddy et al., 2019). Alistarh et al. (2018) proposed a different scheme, where the state of workers, *i.e.,* their gradients and particular estimate sequences, is kept over time, which is used to update the set of good workers at each iteration. This technique might be useful when Byzantines send random updates. However, the server has to keep track of the history of updates by each individual user, which is not practical in large-scale systems. Blanchard et al. (2017) proposed Krum, which is a distance-based gradient aggregation scheme over $L_2$. El Mhamdi et al. (2018) showed that distance-based schemes are vulnerable to *leeway* attacks and proposed Bulyan, which applies an aggregation rule such as Krum iteratively to reject a number of Byzantines followed by a variant of coordinate-wise trimmed mean. Karimireddy et al. (2021) proposed using momentum to defend against time-coupled attacks in an iid setting with a fixed set of Byzantines, which is a different setting compared to our work. While most of the existing results focus on homogeneous data over machines, *i.e.,* iid settings, robust aggregation schemes have been proposed in non-iid settings (Pillutla et al., 2022; Li et al., 2019; Karimireddy et al., 2022). Pillutla et al. (2022) proposed using approximate geometric median of local weights using Smoothed Weiszfeld algorithm. Li et al. (2019) proposed a model aggregation scheme by modifying the original optimization problem and adding an $L_1$ penalty term. Karimireddy et al. (2022) showed that, in non-iid settings, Krum's selection is biased toward certain workers and proposed a method based on resampling to homogenize gradients followed by applying existing aggregation rules. Recently, it has been shown that well-known Byzantine-resilient gradient aggregation schemes are vulnerable to informed and tailored attacks (Fang et al., 2020; Xie et al., 2020b). In this paper, we propose a novel and efficient aggregation scheme, MixTailor, which makes the design of successful attack strategies extremely difficult, if not impossible, for an informed and powerful adversary.

Allen-Zhu et al. (2021) proposed a method where the server keeps track of the history of updates by each individual user. Such additional memory is not required for MixTailor. Recently, Gorbunov et al. (2022) and Zhu et al. (2022) proposed robust methods under bounded global Hessian variance and local Hessian variance, and strong convexity of the loss function, respectively. Such assumptions are not required for MixTailor.

**Robust mean estimation.** The problem of robust mean estimation of a high-dimensional and multi-variate Gaussian distribution has been studied in (Huber, 1964; 2011; Lai et al., 2016; Diakonikolas et al., 2019; Data

& Diggavi, 2021), where unlike corrupted samples, correct samples are evenly distributed in all directions. We note that such strong assumptions do not hold in typical machine learning problems in practice.

**Game theory.** Nash (1950) introduced the notion of mixed strategy in game theory. Unlike game theory, in our setting, the agents, *i.e.,* the server and adversary do not have a complete knowledge of their profiles and payoffs.

***Notation:*** *We use $\mathbb{E}[\cdot]$ to denote the expectation and $\|\cdot\|$ to represent the Euclidean norm of a vector. We use lower-case bold font to denote vectors. Sets and scalars are represented by calligraphic and standard fonts, respectively. We use $[n]$ to denote $\{1, \cdots, n\}$ for an integer $n$.*

## 2 Gradient aggregation, informed adversary, and tailored attacks

Let $\mathbf{w} \in \mathbb{R}^d$ denote a high-dimensional machine learning model. We consider the optimization problem

$$\min_{\mathbf{w} \in \mathbb{R}^d} F(\mathbf{w}) = \frac{1}{n} \sum_{i=1}^{n} F_i(\mathbf{w}) \tag{1}$$

where $F_i : \mathbb{R}^d \to \mathbb{R}$ can be 1) a finite sum representing empirical risk of worker $i$ or 2) $F_i(\mathbf{w}) = \mathbb{E}_{\mathbf{z} \sim \mathcal{D}_i}[\ell(\mathbf{w}; \mathbf{z})]$ in an online setting where $\mathcal{D}_i$ and $\ell(\mathbf{w}; \mathbf{z})$ denote the data distribution of worker $i$ and the loss of model $\mathbf{w}$ on example $\mathbf{z}$, respectively. In federated learning settings, each worker has its own local data distribution, which models *e.g.,* mobile users from diverse geographical regions with diverse socio-economical status (Kairouz et al., 2021).

At iteration $t$, a *good* worker $i$ computes and sends its stochastic gradient $\mathbf{g}_i(\mathbf{w}_t)$ with $\mathbb{E}_{\mathcal{D}_i}[\mathbf{g}_i(\mathbf{w}_t)] = \nabla F_i(\mathbf{w}_t)$. A server aggregates the stochastic gradients following a particular gradient aggregation rule AGG. Then the server broadcasts the updated model $\mathbf{w}_{t+1}$ to all workers. A *Byzantine* worker returns an arbitrary vector such that the basic gradient averaging converges to an ineffective model even if it converges. Byzantine workers may collude and may be omniscient, *i.e.,* they are controlled by an informed adversary with perfect knowledge of the state of the server, prefect knowledge of good workers and transferred data over the network (El Mhamdi et al., 2018; Fang et al., 2020; Xie et al., 2020b). State refers to data and code. The adversary does not have access to the random seed generator at the server. Our model of informed adversary is described in the following.

### 2.1 Informed adversary

We assume an informed adversary has access to the local data stored in $f$ compromised (Byzantine) workers. Note that the adversary controls the output (*i.e.,* computed gradients) of Byzantine workers in all iterations. Those Byzantine workers may output any vector at any step of training, possibly tailor their attacks to corrupt training. Byzantine workers may collude. The adversary cannot control the output of good workers. However, an (unomniscient) adversary may be able to inspect the local data or the output of good workers. On the other hand, an informed adversary, has full knowledge of the local data or the output of all good workers. More importantly, an informed adversary may know the set of aggregation rules that the server applies throughout training. Nevertheless, if the set contains more than one rule, the adversary does not know the random choice of a rule made by the aggregator at a particular instance.[2]

Byzantine workers can optimize their attacks based on gradients sent by good workers and the server's aggregation rule such that the output of the aggregation rule leads to an ineffective model even if it converges. It is shown that well-known Byzantine-resilient aggregation rules with a deterministic structure are vulnerable to such tailored attacks (Fang et al., 2020; Xie et al., 2020b).

---

[2]The exact rule will be determined at the time of aggregation after the updates are received. We assume the server has access to a source of entropy or a secure seed to generate a random number at each iteration, which is a mild assumption (common in cryptography).

## 2.2 Knowledge of the server

We assume that the server knows an upper bound on the number of Byzantine workers denoted by $f$ and that $n \geq 2f + 1$, which is a common assumption in the literature (Blanchard et al., 2017; El Mhamdi et al., 2018; Alistarh et al., 2018; Rajput et al., 2019; Karimireddy et al., 2022).

Suppose there is no Byzantine worker, *i.e.*, $f = 0$. At iteration $t$, good workers compute $\{\mathbf{g}_i(\mathbf{w}_t)\}$. The update rule is given by

$$\mathbf{w}_{t+1} = \mathbf{w}_t - \eta_t \text{AGG}(\{\mathbf{g}_i(\mathbf{w}_t)\}_{i=1}^n)$$

where $\eta_t$ is the learning rate at step $t$ and AGG is an aggregation rule at the server.

**Remark 1.** *To improve communication efficiency, user $i$ may opt to update its copy of model locally for $\tau$ iterations using its own local data and output $\mathbf{w}_t^i$. Then the server aggregates local models, updates the global model $\mathbf{w}_{t+1} = \text{AGG}(\{\mathbf{w}_t^i\}_{i=1}^n)$, and broadcasts the updated model. In this work, we focus on gradient aggregation following the robust aggregation literature. Note that, to improve communication efficiency, a number of efficient gradient compression schemes have been proposed (Alistarh et al., 2017; Faghri et al., 2020; Ramezani-Kebrya et al., 2021). Furthermore, optimizing over $\tau$ is a challenging problem and, to the best of our knowledge, there are very special problems for which local SGD is provably shown to outperform minibatch SGD. Finally, MixTailor is a plug and play scheme, which is compatible with local updating and fine tuning tricks to further improve communication efficiency and fairness.*

## 2.3 Tailored attacks against a given aggregation rule

A tailored attack is designed with prior knowledge of the robust aggregation rule AGG used by the server. Without loss of generality, we assume an adversary controls the first $f$ workers. Let $\mathbf{g} = \text{AGG}(\mathbf{g}_1, \cdots, \mathbf{g}_n)$ denote the aggregated gradient under *no attack*. The Byzantines collude and modify their updates such that the aggregated gradient becomes

$$\mathbf{g}' = \text{AGG}(\mathbf{g}_1', \cdots, \mathbf{g}_f', \mathbf{g}_{f+1}, \cdots, \mathbf{g}_n).$$

A tailored attack is an attack towards the inverse of the direction of $\mathbf{g}$ without attacks:[3]

$$\max_{\mathbf{g}_1', \cdots, \mathbf{g}_f', \lambda} \lambda$$
$$\text{subject to } \mathbf{g}' = \text{AGG}(\mathbf{g}_1', \cdots, \mathbf{g}_f', \mathbf{g}_{f+1}, \cdots, \mathbf{g}_n),$$
$$\mathbf{g}' = -\lambda \mathbf{g}.$$

If feasible, this attack moves the model toward a local *maxima* of our original objective $F(\mathbf{w})$. This attack requires the adversary to have access to the aggregation rule and gradients of all good workers. In Appendix A.1, we extend our consideration to suboptimal tailored attacks, tailored attacks under partial knowledge, and model-based tailored attacks, where the server aggregates the local models instead of gradients.

## 3 Mixed gradient aggregation

It is shown that an informed adversary can efficiently and successfully attack standard robust aggregation rules such as Krum, TrimmedMean, and comed. In particular, Fang et al. (2020); Xie et al. (2020b) found nearly optimal attacks, which are optimized to circumvent aggregation rules with a *deterministic* structure by exploiting the sufficiently large variance of stochastic gradients throughout training deep neural networks. Randomization is the principal way to decrease the degree by which the attacker is informed and thus ensure some level of security.

We propose that, at each iteration, the server draws a robust aggregation rule from a set $M$ computationally-efficient (robust) aggregation rules uniformly at random. We argue that such randomization makes the design

---

[3]This tailored attack is shown to be sufficient against several aggregation rules (Fang et al., 2020); however, it is not necessarily an optimal attack.

of successful and tailored attack strategies extremely difficult, if not impossible, even if an informed adversary has perfect knowledge of the pool of aggregation rules. The specific pool we used for MixTailor is described in Section 5. We should emphasize that our pool is not limited to those aggregation rules that are developed so far. This makes it open for simple extensions by adding new strategies, but the essential protection from randomization remains.

Intuitively, MixTailor creates sufficient uncertainty for the adversary and increases computational complexity of designing tailored attacks, which are guaranteed to corrupt training. To see how MixTailor provides robustness in practice, consider the standard threat model in Byzantine robustness literature: The attack method is decided in advance and the server applies an aggregation strategy to counter this attack. This is favorable for deterministic aggregations such as Krum and comed, but is hardly realistic, as after some time the attacker will find out what the aggregation rule is and use a proper attack accordingly.

An alternative threat model is the aggregation method is known in advance and the attacker applies an attack that is tailored to degrade the chosen aggregation rule. This is a realistic case. To counter it, we introduce MixTailor. By introducing randomization in the aggregation method, we assume we can withhold the knowledge of the exact aggregation rule used in each iteration from the attacker but the attacker can still know the set of aggregation rules in the pool. As we prove, randomization is a principled approach to limit the capability of an attacker to launch tailored attacks in every iteration.

We propose to use a randomized aggregation rule with $M$ candidate rules where $\text{AGG}_m$ is selected with probability $1/M$ such that an informed adversary cannot take advantage of knowing the exact aggregation rule to design an effective attack. Formally, let $V_i(\mathbf{w}) = \mathbf{g}_i(\mathbf{w}, \mathbf{z}) \in \mathbb{R}^d$ be independent random vectors for $i \in [n]$.[4] Let $G(\mathbf{w}, \mathbf{z})$ denote a random function that captures randomness w.r.t. both an honest node $i$ drawn uniformly at random and also an example $\mathbf{z} \sim \mathcal{D}_i$ of that node such that $\mathbb{E}[G(\mathbf{w}, \mathbf{z})] = \nabla F(\mathbf{w})$. Let $\widetilde{\text{AGG}}$ denote a random aggregation rule, which selects a rule from $\{\text{AGG}_1, \cdots, \text{AGG}_M\}$ uniformly at random. Let $\mathcal{B} = \{B_1, \cdots, B_f\}$ denote arbitrary Byzantine gradients, possibly dependent on $V_i$'s. We note that the indices of Byzantines may change over training iterations.

The output of MixTailor algorithm is given by

$$U(\mathbf{w}) = \widetilde{\text{AGG}}(V_1(\mathbf{w}), \cdots, B_1, \cdots, B_f, \cdots, V_n(\mathbf{w})) \tag{2}$$

where $\widetilde{\text{AGG}} = \text{AGG}_m$ with probability $1/M$.

In the following, we define a general robustness definition, which leads to almost sure convergence guarantees to a local minimum of $F$ in (1), which is equivalent to being immune to training-time attacks. Note that our definition covers a general non-iid setting, a general mixed strategy with arbitrary set of candidate robust aggregation rules, and both omniscient and unomniscient adversary models.

**Definition 1.** *Let $\mathbf{w} \in \mathbb{R}^d$. Let $V_i(\mathbf{w}) = \mathbf{g}_i(\mathbf{w}, \mathbf{z}) \in \mathbb{R}^d$ be independent random vectors for $i \in [n]$. Let $G(\mathbf{w}, \mathbf{z})$ denote a random function that captures randomness w.r.t. both an honest node $i$ drawn uniformly at random and also an example $\mathbf{z} \sim \mathcal{D}_i$ of that node such that $\mathbb{E}[G(\mathbf{w}, \mathbf{z})] = \nabla F(\mathbf{w})$. Let $\widetilde{\text{AGG}}$ denote a mixed aggregation rule, which selects a rule from $\{\text{AGG}_1, \cdots, \text{AGG}_M\}$ uniformly at random. Let $\mathcal{B} = \{B_1, \cdots, B_f\}$ denote arbitrary Byzantine gradients, possibly dependent on $V_i$'s.*

*A mixed aggregation rule $\widetilde{\text{AGG}}$ is Byzantine-resilient if $U(\mathbf{w})$ satisfies $\mathbb{E}[U(\mathbf{w})]^\top \nabla F(\mathbf{w}) > 0$ and $\mathbb{E}[\|U(\mathbf{w})\|^r] \leq K_r \mathbb{E}[\|G(\mathbf{w}, \mathbf{z})\|^r]$ for $r = 2, 3, 4$ and some constant $K_r$. Note that the expectation is w.r.t. the randomness in both sampling and aggregation.*

The analysis of computational complexity of MixTailor is discussed in Appendix A.2.

---

[4] In the following, we remove the index $t$ for simplicity.

## 4    Theoretical guarantees

We first provide a sufficient condition to guarantee that MixTailor algorithm is Byzantine-resilient according to Definition 1. Proofs are in appendices. Let

$$U_m(\mathbf{w}) = \text{AGG}_m(V_1(\mathbf{w}), \cdots, B_1, \cdots, B_f, \cdots, V_n(\mathbf{w}))$$

denote the output of $\text{AGG}_m$ for $m \in [M]$.

**Proposition 1.** *Let $\mathbf{w} \in \Omega$ and $0 < q < M$. Let $G(\mathbf{w}, \mathbf{z})$ denote a random function that captures randomness w.r.t. both an honest node $i$ drawn uniformly at random and also an example $\mathbf{z} \sim \mathcal{D}_i$ of that node such that $\mathbb{E}[G(\mathbf{w}, \mathbf{z})] = \nabla F(\mathbf{w})$. Let $L > 0$ denote the Lipschitz parameter of $F$. Let $\mathcal{B}$ denote an attack against $q$ aggregation rules such that $\mathbb{E}[U_i(\hat{\mathbf{w}})]^\top \nabla F(\hat{\mathbf{w}}) < 0$ for some $\hat{\mathbf{w}}$ and $i \in [q]$. Suppose that aggregation rules $\text{AGG}_{m'}$'s are resilient against this attack, i.e.,*

$$\mathbb{E}[U_{m'}(\mathbf{w})]^\top \nabla F(\mathbf{w}) \geq \beta_{m'} > 0$$

*and $\mathbb{E}[\|U(\mathbf{w})\|^r] \leq K_r \mathbb{E}[\|G(\mathbf{w}, \mathbf{z})\|^r]$ with $U(\mathbf{w})$ in (2) for $r = 2, 3, 4$, some constant $K_r$, and $m' \in \{q + 1, \cdots, M\}$. Suppose that $M$ is large enough such that*

$$\frac{M}{q} > 1 + \frac{\lambda L}{\min_{m'} \beta_{m'}}$$

*where $\lambda = \max_{i \in [q]} \sup_{\mathbf{w} \in \Omega} \|\mathbb{E}[U_i(\mathbf{w})]\|$, then the mixed aggregation rule $\widetilde{\text{AGG}}$ is resilient against any such $\mathcal{B}$.*

*Proof.* See Appendix A.3.    □

Proposition 1 shows resilience of the mixed aggregation rule when only a subset of rules are resilient against an attack no matter how the attack is designed (it could be computationally expensive).

**Remark 2.** *There are possibly tailored attacks against any individual aggregation rule. On the other hand, all aggregation rules are not vulnerable to the same attack. In sum, robustness is achieved as long as we have a sufficiently diverse set of aggregation rules in our pool. In (Fang et al., 2020, Theorem 1), an upper bound is established on the norm of the attack vector that is tailored against Krum. In (Xie et al., 2020b, Theorem 1), a lower bound is established on the norm of the attack that is tailored against comed. Theoretical results are consistent with the attacks developed empirically in (Fang et al., 2020; Xie et al., 2020b) and confirm that Krum and comed are indeed vulnerable to different types of attacks, so they are diverse w.r.t. their vulnerabilities. We emphasize that the key element that provides robustness is randomization.*

**Remark 3.** *Let $\hat{\mathbf{w}} \in \mathbb{R}^d$. To fail the conditions specified in Proposition 1, an adversary should have sufficient computational resources to find an attack (if exists) such that $\mathbb{E}[U_i(\hat{\mathbf{w}})]^\top \nabla F(\hat{\mathbf{w}}) < 0$ for $i \in [q]$ where $q$ should be large enough. Suppose that the adversary has sufficient random samples from each honest client to compute the expectation over the output of an aggregation rule $\text{AGG}_i$ and has access to an accurate estimate of $\nabla F(\hat{\mathbf{w}})$. An aggregation $\text{AGG}_i$ is typically a nonconvex function of the attack $\mathcal{B}$ in Proposition 1. Instead of designing an optimal attack, suppose that the adversary plans to verify an attack, which is a computationally simpler problem. By verification, we mean computing the output of $\text{AGG}_i$ under an attack $\mathcal{B}$ and computing the sign of $\mathbb{E}[U_i(\hat{\mathbf{w}})]^\top \nabla F(\hat{\mathbf{w}})$. The verification runtime increases monotonically as $q$ increases. We note that due to nonconvexity of baseline aggregation rules such as comed and Krum, we are unaware of any polynomial time algorithm with provable guarantees to efficiently corrupt multiple aggregation rules at the same time.*

In Appendix A.4, we define an optimal training-time attack and discuss an alternative attack design based on a min-max problem.

### 4.1    Attack complexity

Unlike hyperparameter-based randomization techniques such as sub-sampling, MixTailor provides randomness in the *structure of aggregation rules*, which makes it impossible for the attacker to *control the optimization*

*trajectory.* Hyperparameter-based randomization techniques as sub-sampling can also improve robustness by some extent, however, the adversary can still design a successful attack by focusing *on the specific aggregation structure.* The adversary can do so for example by mimicking the subsampling procedure to fail it.

Formally, suppose that The set of $M$ aggregators used by the server is denoted by $\mathcal{A} = \{\mathrm{AGG}_1, \mathrm{AGG}_2, \ldots, \mathrm{AGG}_M\}$. We note that each aggregation rule $\mathrm{AGG}_i$ is either deterministic or has some hyperparameters which can be set randomly such as sub-sampling parameters. For each $\mathrm{AGG}_i$, we define attack complexity as follows:

Let $T_i(n, f, d, \epsilon)$ denote the *number of elementary operations* an informed adversary requires to design a tailored attack in terms of solving the optimization problem in Section 2.3 with precision $\epsilon$ for satisfying constraints such that $\epsilon = \arg\cos\left(\frac{-\mathbf{g}^\top \mathbf{g}'}{\|\mathbf{g}'\|\|\mathbf{g}\|}\right) = \pi - \arg\cos\left(\frac{\mathbf{g}^\top \mathbf{g}'}{\|\mathbf{g}'\|\|\mathbf{g}\|}\right)$. For a given $\mathrm{AGG}_i$, the number of elementary operations increases to achieve smaller values of $\epsilon$, which amounts to optimizing more effective attacks. We note that all realizations of aggregations with a random hyperparameter but the same structure, for example Krum with various sub-sampling parameters, have the same attack complexity. The attack complexity for MixTailor is $\Omega(\sum_{i=1}^{M} T_i(n, f, d, \epsilon))$, which monotonically increases by $M$. To see this, assume there exists an attacker with lower complexity, then the attacker fails to break at least one of the aggregators. Note that precise expressions of $T_i$'s, i.e., the exact numbers of elementary operations depend on the specific problem to solve (dataset, loss, architecture, etc), and the hyperparameters to chosen (for example aggregators used for the selection and aggregation phases of Bulyan), the optimization method the attacker uses for designing an attack, and the implementation details (code efficiency).

## 4.2 Generalized Krum

In the following, we develop a lower bound on $\mathbb{E}[U_m(\mathbf{w})]^\top \nabla F(\mathbf{w})$ when $\mathrm{AGG}_m$ is a generalized version of Krum. Let $\mathcal{G}$ and $\mathcal{B}$ denote the set of good and Byzantine workers, respectively. Let $\mathcal{N}_g(i)$ and $\mathcal{N}_b(i)$ denote the set good and Byzantine workers among $n - f - 2$ closest values to the gradient (model update) of worker $i$. We consider a generalized version of Krum where $\mathrm{AGG}_m$ selects a worker that minimizes this score function: Let $p \geq 1$ specifies a particular norm. The generalized Krum selects worker

$$i^* = \arg\min_{i \in [n]} \sum_{j \in \mathcal{N}_g(i) \cup \mathcal{N}_b(i)} \|G_i - G_j\|_p^2 \tag{3}$$

where $G_i$ is the update from worker $i$. Note that $G_i$ can be either $V_i(\mathbf{w})$ or $B_i$ depending on whether worker $i$ is good or Byzantine. We drop $\mathbf{w}$ and subscript $m$ for notational simplicity. We first find upper bounds on $\|\mathbb{E}[U(\mathbf{w})] - \nabla F(\mathbf{w})\|_2^2$ and $\mathbb{E}[\|U(\mathbf{w})\|_2^r]$ for $r = 2, 3, 4$.

**Theorem 1.** *Let $\mathbf{w} \in \mathbb{R}^d$ and $p \geq 1$. Suppose $\mathbb{E}[\|V_i(\mathbf{w}) - \nabla F(\mathbf{w})\|_2^2] \leq \sigma^2$ for all good workers $V_i(\mathbf{w}) \in \mathcal{G}$. The output of $\mathrm{AGG}$ in (3) guarantees:*

$$\|\mathbb{E}[U(\mathbf{w})] - \nabla F(\mathbf{w})\|_2^2 \leq 2\sigma^2 \big(1 + \Lambda(n, f, d, p)\big)$$

*where $\Lambda(n, f, d, p) = d^{\frac{\max\{p,2\} - \min\{p,2\}}{p}} C(n, f)$ and $C(n, f) = 1 + \frac{2f}{n - 2f - 2}$. In addition, for $r = 2, 3, 4$, there is a constant $C$ such that $\mathbb{E}[\|U(\mathbf{w})\|_2^r] \leq C\mathbb{E}[\|G(\mathbf{w}, \mathbf{z})\|_2^r]$.*

*Proof.* See Appendix A.5. □

**Remark 4.** *The bounds available in (Blanchard et al., 2017; Karimireddy et al., 2022) can be considered as special cases of our bounds for the case of $p = 2$. Our analysis is tighter than those bounds for this special case.*

## 4.3 Non-iid setting

We now consider a non-iid setting assuming bounded inter-client gradient variance, which is a common assumption in federated learning literature (Kairouz et al., 2021, Sections 3.2.1 and 3.2.2). We find an upper bound on $\|\mathbb{E}[U(\mathbf{w})] - \nabla F(\mathbf{w})\|_2^2$ for the generalized Krum in (3). Our assumption is as follows:

**Assumption 1.** *Let $\mathbb{E}_{\mathcal{D}_i}[V_i(\mathbf{w})] = \mathbf{g}_i(\mathbf{w})$. For all good workers $i \in \mathcal{G}$ and all $\mathbf{w}$, we assume*

$$\mathbb{E}_{\mathcal{D}_i}[\|V_i(\mathbf{w}) - \mathbf{g}_i(\mathbf{w})\|_2^2] \leq \sigma^2,$$

$$\frac{1}{n-f} \sum_{i=1}^{n-f} \|\mathbf{g}_i(\mathbf{w}) - \nabla F(\mathbf{w})\|_2^2 \leq \Delta^2.$$

Recall that $G(\mathbf{w}, \mathbf{z})$ denotes a random function that captures randomness w.r.t. both an honest node $i$ drawn uniformly at random and also an example $\mathbf{z} \sim \mathcal{D}_i$ of that node such that $\mathbb{E}[G(\mathbf{w}, \mathbf{z})] = \nabla F(\mathbf{w})$. The following assumption, which bounds higher-order moments of the gradients of good workers, is needed to prove almost sure convergence (Bottou, 1998; Blanchard et al., 2017; Karimireddy et al., 2022).

**Assumption 2.** $\mathbb{E}_{\mathcal{D}_i}[\|V_i(\mathbf{w})\|_2^r] \leq K_{r,i}\mathbb{E}[\|G(\mathbf{w}, \mathbf{z})\|_2^r]$ *for $r = 2, 3, 4$, $i \in \mathcal{G}$, and some constant $K_{r,i}$.*

**Theorem 2.** *Let $\mathbf{w} \in \mathbb{R}^d$ and $p \geq 1$. Under Assumption 1, the output of AGG in (3) guarantees:*

$$\|\mathbb{E}[U(\mathbf{w})] - \nabla F(\mathbf{w})\|_2^2 \leq C_1 + C_2\Lambda(n, f, d, p).$$

*where $C_1 = 6\sigma^2 + 2\left(n - f + 3 + \frac{2(n-f)}{n-2f-2}\right)\Delta^2$ and $C_2 = 4\sigma^2 + 8(n-f)\Delta^2$. In addition, under Assumption 2 and for $r = 2, 3, 4$, there is a constant $C$ such that $\mathbb{E}[\|U(\mathbf{w})\|_2^r] \leq C\mathbb{E}[\|G(\mathbf{w}, \mathbf{z})\|_2^r]$.*

*Proof.* See Appendix A.6. $\qquad\qquad\qquad\qquad\qquad\qquad\qquad\qquad\qquad\qquad\qquad\qquad\qquad\quad\square$

Note that our bound recovers the results in Theorem 1 in the special case of homogeneous data. Substituting $\Delta = 0$, we note that the constant term in Theorem 1 is slightly smaller than that in Theorem 2.

**Remark 5.** *Note that $C_1$ and $C_2$ are monotonically increasing with $n$, which is due to data heterogeneity. Even without Byzantines, we can establish a lower bound on the worst-case variance of a good worker that grows with $n$.*

Finally, for both iid and non-iid settings and a general nonconvex loss function, we can establish almost sure convergence ($\nabla F(\mathbf{w}_t) \to 0$ a.s.) of the output of AGG in (3) along the lines of (Fisk, 1965; Métivier, 1982; Bottou, 1998). The following theorem statement is for the non-iid setting.

**Theorem 3.** *Let $\mathbf{w} \in \mathbb{R}^d$ and $p \geq 1$. Let $n > 1$ and $f \geq 0$ denote integers with $n > 2f + 2$. Let $C_{0,r}, C_{1,r}$ denote some constants for $r = 2, 3, 4$, $F(\mathbf{w}) \geq 0$ denote a possibly nonconex and three times differentiable function[5] with continuous derivatives, and $G(\mathbf{w}, \mathbf{z})$ denote a random vector such that $\mathbb{E}[G(\mathbf{w}, \mathbf{z})] = \nabla F(\mathbf{w})$ and $\mathbb{E}[\|G(\mathbf{w}, \mathbf{z})\|_2^r] \leq C_{0,r} + C_{1,r}\|\mathbf{w}\|_2^r$ for $r = 2, 3, 4$. Suppose that the generalized Krum algorithm with AGG in (3) is executed with a learning rate schedule $\{\eta_t\}$, which satisfies $\sum_t \eta_t = \infty$ and $\sum_t \eta_t^2 < \infty$. Suppose there exists $\beta > 0$, $R > 0$, and $0 \leq \theta < \pi/2 - \sup_{\|\mathbf{w}\|_2^2 \geq R} \alpha$ such that $\inf_{\|\mathbf{w}\|_2^2 \geq R} \|\mathbb{E}[U(\mathbf{w})]\|_2^2 + \inf_{\|\mathbf{w}\|_2^2 \geq R} \|\nabla F(\mathbf{w})\|_2^2 - C_1 - C_2\Lambda(n, f, d, p) \geq \beta$ and*

$$\inf_{\|\mathbf{w}\|_2^2 \geq R} \frac{\mathbf{w}^\top \nabla F(\mathbf{w})}{\|\mathbf{w}\|_2 \|\nabla F(\mathbf{w})\|_2} \geq \cos(\theta)$$

*where*

$$\alpha = \arccos\left(\frac{\beta}{2\|\mathbb{E}[U(\mathbf{w})]\|_2 \|\nabla F(\mathbf{w})\|_2}\right).$$

*Then the sequence of gradients $\{\nabla F(\mathbf{w}_t)\}$ converges to zero almost surely.*

*Proof.* See Appendix A.7. $\qquad\qquad\qquad\qquad\qquad\qquad\qquad\qquad\qquad\qquad\qquad\qquad\qquad\quad\square$

In Proposition 1 and Theorem 3, we find conditions, for example an upper bound on the number of failed aggregation rules under an attack, under which MixTailor is guaranteed to *converge to an empirical risk minimizer of the original objective of honest workers*, i.e., converge to an effective model.

---

[5]It can be the true risk in an online setting.

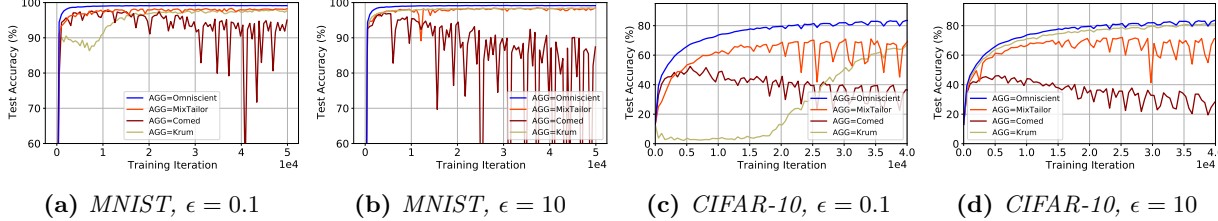

**(a)** *MNIST, $\epsilon = 0.1$*  **(b)** *MNIST, $\epsilon = 10$*  **(c)** *CIFAR-10, $\epsilon = 0.1$*  **(d)** *CIFAR-10, $\epsilon = 10$*

**Figure 1:** ***Test accuracy on MNIST and CIFAR-10 under iid setting.*** *MixTailor outperforms individual aggregators in terms of robustness. Tailored attacks ($\epsilon = 0.1, 10$) are applied at each iteration. There are $n = 12$ total workers including $f = 2$ Byzantine workers. The dataset is randomly and equally partitioned among workers. The omniscient aggregator receives and averages 10 honest gradients at each iteration. For MixTailor, we randomly select an aggregator from the pool of 64 aggregators at each iteration.*

## 5 Experimental evaluation

In this section, we evaluate the resilience of MixTailor against tailored attacks. We construct a pool of aggregators based on 4 robust aggregation rules: comed (Yin et al., 2018), Krum (Blanchard et al., 2017), an efficient implementation of geometric median (Pillutla et al., 2022), and Bulyan (El Mhamdi et al., 2018). Each Bulyan aggregator uses a different aggregator from Krum, average, geometric median, and comed for either the selection phase or in the aggregation phase. For each class, we generate 16 aggregators, each with a randomly generated $\ell^p$ norm from one to 16. MixTailor selects one aggregator from the entire pool of 64 aggregators uniformly at random at each iteration. [6] We compare MixTailor with the following baselines: omniscient, which receives and averages all honest gradients at each iteration, vanilla comed, and vanilla Krum. Our results for variations of MixTailor under different pools along with additional experimental results are provided in Appendix A.8. In particular, we show the performance of modified versions of MixTailor under tailored attacks and MixTailor under "A Little" attack (Baruch et al., 2019).

We simulate training with 12 total workers, where 2 workers are compromised by an informed Byzantine workers sending tailored attacks. We train a CNN model on MNIST (LeCun et al., 1998) and CIFAR-10 (Krizhevsky) under both iid and non-iid settings. The details of the model and training hyper-parameters are provided in Appendix A.8. In the iid settings (Fig. 1), the dataset is shuffled and equally partitioned among workers. In the non-iid setting (Fig. 3), the dataset is first sorted by labels and then partitioned among workers such that each good worker computes its local gradient on particular examples corresponding to a label. This creates statistical heterogeneity across good workers. In both settings, the informed adversary has access to the gradients of honest workers. Our PyTorch code will be made publicly available (Paszke et al., 2019).

We consider tailored attacks as described in (Fang et al., 2020; Xie et al., 2020b). The adversary computes the average of correct gradients, scales the average with a parameter $\epsilon$, and has the Byzantine workers send back scaled gradients towards the inverse of the direction along which the global model would change without attacks. Since the adversary does not know the exact rule in the randomized case in each iteration, we use $\epsilon$'s that are proposed in (Fang et al., 2020; Xie et al., 2020b) for our baseline deterministic rules. A small $\epsilon$ corrupts Krum, while a large one corrupts comed.

**Consistent robustness across the course of training.** Our randomized scheme successfully decreases the capability of the adversary to launch tailored attacks. Fig. 1a and Fig. 1b show test accuracy when we train on MNIST under tailored attacks proposed by (Fang et al., 2020; Xie et al., 2020b). Fig. 2 shows that a setting where Krum fails while MixTailor is able to defend the attacks. The reason that MixTailor is able to defend is using aggregators that are able to defend against this attack such as comed and geometric median. We note that MixTailor consistently defends when vanilla Krum and comed fail. In addition, compared with Krum and comed, MixTailor has much less fluctuations in terms of test accuracy across the course of training.

---

[6]To ensure that the performance of MixTailor is not dominated by a single aggregation rule, in Appendix A.8, we show the results when we remove a class of aggregation rule (each with 16 aggregators) from MixTailor pool. We observe that MixTailor with a smaller pool performs roughly the same.

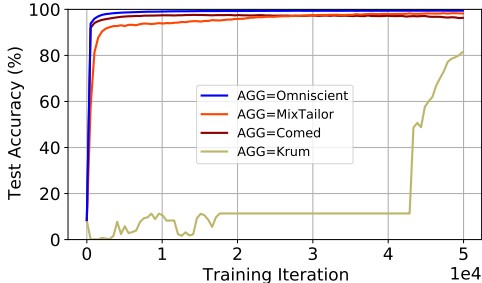 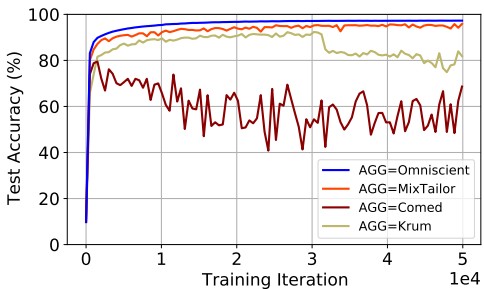

**Figure 2:** ***Krum fails.*** *Test accuracy on MNIST with $\epsilon = 0.2$ for the tailored attack. We set $n = 12$ and $f = 2$. The batch size is set to 128. The dataset is randomly and equally partitioned among workers. Omniscient receives and averages 10 honest gradients at each iteration.*

**Figure 3:** ***Test accuracy on MNIST under non-iid setting.*** *MixTailor is robust to Byzantine workers in the heterogeneous setting ($\epsilon = 0.1$). The dataset is partitioned by labels such that each worker holds samples for a single digit. The rest of the setup is similar to Fig. 1.*

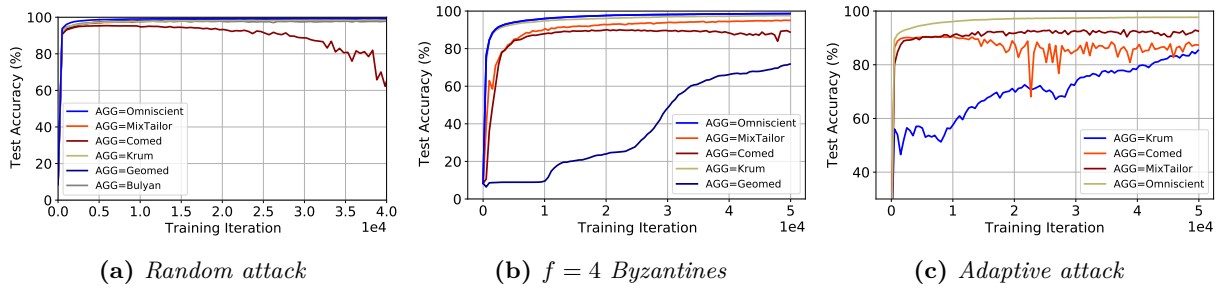

**(a)** *Random attack*  **(b)** $f = 4$ *Byzantines*  **(c)** *Adaptive attack*

**Figure 4:** *Test accuracy on MNIST with random attack (left), $f = 4$ Byzantines under $\epsilon = 10$ (middle), and adaptive attack (right). MixTailor is robust to random and adaptive attacks and always outperforms the worst aggregator. The rest of the setup is similar to Fig. 1.*

MixTailor reaches within 1% of the accuracy of the omniscient aggregator, under $\epsilon = 0.1$ and $\epsilon = 10$ attacks, respectively.

**There is a free lunch!** Note that the robustness of MixTailor comes at no additional computation cost. As discussed in Section 3, the per-step computation cost of MixTailor is on par with deterministic aggregation rules. In addition, MixTailor does not impose any additional communication cost. Our comparison in terms of the number of training iterations is independent of a particular distributed setup.

**Deterministic methods are sensitive to the attack's parameters.** We note that, unlike Krum and comend, the performance of MixTailor is stable across large and small $\epsilon$'s. Fig. 1c and Fig. 1d show test accuracy when we train on CIFAR-10. The attack with small $\epsilon$ successfully corrupts Krum in the beginning of training. Note that we have not optimized $\epsilon$ and opted to use those proposed by Xie et al. (2020b). Surprisingly, we observe that comed fails under both small and large $\epsilon$'s on CIFAR-10. We also observed that comed is also unstable to the choice of hyper-parameters. In particular, we noticed that comed does not converge when the learning rate is set to 0.1 even when there are no Byzantines. To the best of our knowledge, this vulnerability has not been reported in the literature. We emphasize that under each attack, MixTailor always outperforms the worst aggregator but there is an aggregator in the pool that outperforms MixTailor.

**MixTailor with resampling in non-iid settings.** Finally, we note that MixTailor can be combined with various techniques that are proposed to handle data heterogeneity. In particular, we used resampling before all robust aggregation methods (Karimireddy et al., 2022). Resampling is a simple method which homogenizes the received gradients before aggregating. In particular, in Fig. 3 we show test accuracy when MNIST is partitioned among workers in a non-iid manner. Similar to the iid case, MixTailor shows consistent robustness across the course of training.

**Table 1:** Time per iteration. Computational cost of each aggregator. The aggregation methods use 12 workers. The computational cost is collected on a T4 GPU. The Bulyan aggregator uses Krum for the aggregation phase and FedAvg for the selection phase.

| Aggregator | Time per iteration (us) |
|:---:|:---:|
| Omniscient | 60 |
| MixTailor | 4980 |
| Krum | 2176 |
| Comed | 153 |
| Bulyan | 6700 |

**Comparison with geomed and Bulyan, random-$\epsilon$ attack, and more number of Byzantine workers.** We evaluate MixTailor and other rules against the random attack randomly drawn from the set of a small $\epsilon$ to corrupt Krum, a large one to corrupt comed. Fig. 4a shows that such random attack is not as effective as tailored attacks against any specific rule. For a training-time attack to be effective, it should be applied consistently for some consecutive iterations.The best attack against any deterministic rule is designed deterministically against that rule. Fig. 4b shows the results with 4 Byzantines under $\epsilon = 10$. Due to the structure of Bulyan (it requires $n > 4f + 3$), we had to remove it from the pool of aggregators for MixTailor. We note that geomed is vulnerable to this attack.

We focused on Krum and comed since we are aware of tailored attacks against them (Fang et al., 2020; Xie et al., 2020b). Fig 4b shows that geomed may be vulnerable to such attacks designed for Krum and comed. MixTailor always outperforms the worst aggregator, which is the target of a tailored attack.

**MixTailor under an adaptive attack.** We have considered a stronger and adaptive attacker, which optimizes its attack by enumerating over a set of $\epsilon$'s and selects *the worst $\epsilon$ against the aggregator at every single iteration.* The adversary enumerates among all those $\epsilon$'s and finds out which one is the most effective attack by applying the aggregator (the attacker simulates the server job by applying the aggregator with different $\epsilon$'s and finds the best attack and then outputs the best attack for the server to aggregate). Regarding MixTailor, the attacker selects a random aggregator from the MixTailor's aggregator pool in each iteration and finds the worst epsilon corresponding to this aggregator. The attacker finds *an adaptive* attack by calculating the dot product of the output of the aggregator and the direction of aggregated gradients without attacks when different epsilons are fed into the aggregator. The attacker chooses the epsilon that causes the aggregator to produce the gradient that has the smallest dot product with the true gradient. Note that in order to keep the computational cost of the attack similar to the Comed and Krum baselines, the *adaptive* attack selects an aggregator randomly in each iteration and finds the worst $\epsilon$ with regard to this aggregator. In Fig. 4c, we ran this experiment over MNIST and observe that MixTailor is able to outperform both Krum and Comed. Comed's accuracy changes between 85-87%, Krum's accuracy is 83-85%, and MixTailor's accuracy is 91.80-92.55%. The accuracy of the omniscient aggregator is 97.62-97.68%. The set of epsilons used by the adaptive attacker is 0.1, 0.5, 1, and 10.

**Computational costs.** In Table 1, we empirically provide computational costs for different aggregation rules after running 10 iterations. This table shows the time per iteration for each aggregator used. The average computation cost of MixTailor across the course of training is the average of the costs for candidate rules. As $M$ increases, the average time per iteration for MixTailor increases linearly with the average computation costs of $M$ underlying aggregators.

## 6 Conclusions and future work

To increase computational complexity of designing tailored attacks for an informed adversary, we introduce MixTailor based on randomization of robust aggregation strategies. We provide a sufficient condition to guarantee the robustness of MixTailor based on a generalized notion of Byzantine-resilience in non-iid settings. Under both iid and non-iid data, we establish almost sure convergence guarantees that are both stronger

and more general than those available in the literature. We demonstrate the superiority of MixTailor over deterministic robust aggregation schemes empirically under various attacks and settings. Beyond tailored attacks in (Fang et al., 2020; Xie et al., 2020b), we show the superiourity of MixTailor under a stronger and adaptive attacker, which optimizes its attack at every single iteration.

In this paper, we focus on stationary settings where the maximum gradient norm of the loss across the course of training is sufficiently small such that the effective poison cannot change the accuracy much at a single iteration. Developing defense mechanisms in more challenging settings where an adversary is able to design an effective poison in one iteration is an interesting problem for future work.

Recently, Yang & Bajwa (2019); Peng & Ling (2020); Xie et al. (2020c) proposed Byzantine-resilient schemes for decentralized and asynchronous settings. Extending the structure of MixTailor to those settings is an interesting problem for future work. Secure aggregation rules guarantee some level of input privacy. However, secure aggregation creates additional vulnerabilities and makes defenses more challenging since the server only observes the aggregate of client's updates (Kairouz et al., 2021). Developing efficient and secure protocols to compute MixTailor using, *e.g.,* multi-party computation remains an open problem for future research.

### Acknowledgments

The authors would like to thank Daniel M. Roy, Sadegh Farhadkhani, and our reviewers at Transactions on Machine Learning Research (TMLR) for providing helpful suggestions, which improve the quality of the paper and clarity of presentation. Ramezani-Kebrya was supported by an NSERC Postdoctoral Fellowship. Faghri was supported by an OGS Scholarship. Resources used in preparing this research were provided, in part, by the Province of Ontario, the Government of Canada through CIFAR, and companies sponsoring the Vector Institute.[7]

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

## A  Appendix

### A.1  Tailored attacks

Without loss of generality (W.L.O.G.), we assume an adversary controls the first $f$ workers. Let $\mathbf{g} = \mathrm{AGG}(\mathbf{g}_1, \cdots, \mathbf{g}_n)$ denote the current gradient under *no attack*. The Byzantines collude and modify their updates such that the aggregated gradient becomes

$$\mathbf{g}' = \mathrm{AGG}(\mathbf{g}'_1, \cdots, \mathbf{g}'_f, \mathbf{g}_{f+1}, \cdots, \mathbf{g}_n).$$

A tailored attack is an attack towards the inverse of the direction of $\mathbf{g}$ without attacks:

$$\max_{\mathbf{g}'_1, \cdots, \mathbf{g}'_f, \lambda} \lambda$$
$$\text{subject to } \mathbf{g}' = \mathrm{AGG}(\mathbf{g}'_1, \cdots, \mathbf{g}'_f, \mathbf{g}_{f+1}, \cdots, \mathbf{g}_n),$$
$$\mathbf{g}' = -\lambda \mathbf{g}.$$

If successful, this attack moves the model toward a local *maxima* of our original objective $F(\mathbf{w})$. This attack requires the adversary to have access to the aggregation rule and gradients of all good workers. We now extend our consideration to suboptimal tailored attacks, tailored attacks under partial knowledge, and model-based tailored attacks, where the server aggregates local models instead of gradients.

#### A.1.1  Suboptimal tailored attacks

To reduce computational complexity of the problem of designing successful attacks, we consider the solution to the following problem by restricting the space of optimization variables, which returns a nearly optimal tailored attack (Fang et al., 2020; Xie et al., 2020b):

$$\max_{\mathbf{g}'_1, \lambda} \lambda$$
$$\text{subject to } \mathbf{g}' = \mathrm{AGG}\Big( \underbrace{\mathbf{g}'_1, \mathbf{g}'_1, \cdots, \mathbf{g}'_1}_{f}, \mathbf{g}_{f+1}, \cdots, \mathbf{g}_n \Big),$$
$$\mathbf{g}' = -\lambda \mathbf{g}.$$

#### A.1.2  Tailored attacks under partial knowledge

Now suppose, an unomniscient adversary has access to the models of workers $\mathbf{g}_{f+1}, \cdots, \mathbf{g}_k$. We propose a tailored attack towards the inverse of the direction of $\hat{\mathbf{g}} = \mathrm{AGG}\Big(\mathbf{g}_1, \cdots, \mathbf{g}_k, \underbrace{\sum_{i=1}^{k} \mathbf{g}_i/k, \cdots, \sum_{i=1}^{k} \mathbf{g}_i/k}_{n-k}\Big)$. An

optimal attack is the solution of the following problem:

$$\max_{\mathbf{g}'_1,\cdots,\mathbf{g}'_f,\lambda} \lambda$$

$$\text{subject to } \mathbf{g}' = \text{AGG}\Big(\mathbf{g}'_1,\cdots,\mathbf{g}'_f,\mathbf{g}_{f+1},\cdots,\mathbf{g}_k,\underbrace{\overline{\mathbf{g}},\cdots,\overline{\mathbf{g}}}_{n-k}\Big),$$

$$\mathbf{g}' = -\lambda\hat{\mathbf{g}}.$$

where $\overline{\mathbf{g}} = \sum_{i=1}^{k}\mathbf{g}_i/k$.

A suboptimal attack under partial knowledge is given by a solution to this problem:

$$\max_{\mathbf{g}'_1,\lambda} \lambda$$

$$\text{subject to } \mathbf{g}' = \text{AGG}\Big(\underbrace{\mathbf{g}'_1,\mathbf{g}'_1,\cdots,\mathbf{g}'_1}_{f},\mathbf{g}_{f+1},\cdots,\mathbf{g}_k,\underbrace{\overline{\mathbf{g}},\cdots,\overline{\mathbf{g}}}_{n-k}\Big),$$

$$\mathbf{g}' = -\lambda\hat{\mathbf{g}}.$$

### A.1.3   Model-based tailored attacks

Let $\mathbf{w}_0$ denote the previous weight vector sent by the server. W.L.O.G., we assume an adversary controls the first $f$ workers. Let $\mathbf{w} = \text{AGG}(\mathbf{w}_1,\cdots,\mathbf{w}_n)$ denote the current model under no attack. The Byzantines collude and modify their weights such that the aggregated model becomes

$$\mathbf{w}' = \text{AGG}(\mathbf{w}'_1,\cdots,\mathbf{w}'_f,\mathbf{w}_{f+1},\cdots,\mathbf{w}_n).$$

We consider an attack towards the inverse of the direction along which the global model would change without attacks:

$$\max_{\mathbf{w}'_1,\cdots,\mathbf{w}'_f,\lambda} \lambda$$

$$\text{subject to } \mathbf{w}' = \text{AGG}(\mathbf{w}'_1,\cdots,\mathbf{w}'_f,\mathbf{w}_{f+1},\cdots,\mathbf{w}_n),$$

$$\mathbf{w}' = \mathbf{w}_0 - \lambda\mathbf{s}.$$

where $\mathbf{s} = \mathbf{w} - \mathbf{w}_0$ is the changing direction of global model parameters under no attack. Note that different from (Fang et al., 2020), we consider $\mathbf{w} - \mathbf{w}_0$ instead of $\text{sign}(\mathbf{w} - \mathbf{w}_0)$, and we present an optimal attack, where each Byzantine can inject its own attack independent of other workers.

### A.1.4   Model-based and suboptimal tailored attacks

To reduce computation complexity of the problem of designing successful attacks, we also consider the solution to the following problem by restricting the space of optimization variables, which returns a nearly optimal tailored attack (Fang et al., 2020; Xie et al., 2020b):

$$\max_{\mathbf{w}'_1,\lambda} \lambda$$

$$\text{subject to } \mathbf{w}' = \text{AGG}\Big(\underbrace{\mathbf{w}'_1,\mathbf{w}'_1,\cdots,\mathbf{w}'_1}_{f},\mathbf{w}_{f+1},\cdots,\mathbf{w}_n\Big),$$

$$\mathbf{w}' = \mathbf{w}_0 - \lambda\mathbf{s}.$$

### A.1.5   Model-based tailored attacks under partial knowledge

Now suppose, an unomniscient adversary has access to the models of workers $\mathbf{w}_{f+1},\cdots,\mathbf{w}_k$. We propose a tailored attack towards the inverse of the direction along which the global model would change under

$\hat{\mathbf{w}} = \text{AGG}\Big(\mathbf{w}_1, \cdots, \mathbf{w}_k, \underbrace{\overline{\mathbf{w}}, \cdots, \overline{\mathbf{w}}}_{n-k}\Big)$. An optimal attack is the solution to the following problem:

$$\max_{\mathbf{w}_1', \cdots, \mathbf{w}_f', \lambda} \lambda$$

$$\text{subject to } \mathbf{w}' = \text{AGG}\Big(\mathbf{w}_1', \cdots, \mathbf{w}_f', \mathbf{w}_{f+1}, \cdots, \mathbf{w}_k, \underbrace{\overline{\mathbf{w}}, \cdots, \overline{\mathbf{w}}}_{n-k}\Big),$$

$$\mathbf{w}' = \mathbf{w}_0 - \lambda\hat{\mathbf{s}}$$

where $\hat{\mathbf{s}} = \hat{\mathbf{w}} - \mathbf{w}_0$ and $\overline{\mathbf{w}} = \sum_{i=1}^k \mathbf{w}_i/k$.

A suboptimal attack under partial knowledge is given by a solution to this problem:

$$\max_{\mathbf{w}_1', \lambda} \lambda$$

$$\text{subject to } \mathbf{w}' = \text{AGG}\Big(\underbrace{\mathbf{w}_1', \cdots, \mathbf{w}_1'}_{f}, \mathbf{w}_{f+1}, \cdots, \mathbf{w}_k, \underbrace{\overline{\mathbf{w}}, \cdots, \overline{\mathbf{w}}}_{n-k}\Big),$$

$$\mathbf{w}' = \mathbf{w}_0 - \lambda\hat{\mathbf{s}}.$$

## A.2 Computational complexity

The worst-case computational cost of MixTailor is upper bounded by that of the candidate aggregation rule with the maximum number of elementary operations per iteration. The average computation cost across the course of training is the average of the costs for candidate rules. In particular, the number of operations per iteration for Bulyan with Krum as its aggregation rule is in the order of $O(n^2d)$ (El Mhamdi et al., 2018). The computational cost for coordinate-wise median and an efficient implementation of an approximate geometric median based on Weiszfeld algorithm is $O(nd)$ (Pillutla et al., 2022). Increasing the number of aggregators in the pool does not necessarily increase the computation costs of MixTailor as long as the number of elementary operations per iteration for new aggregators is in the order of those rules that have been already in the pool of aggregators.

## A.3 Proof of Proposition 1

Let $\mathcal{B} = \{B_1, \cdots, B_f\}$ be an attack designed against $\text{AGG}_i$ such that $\mathbb{E}[U_i(\hat{\mathbf{w}})]^\top \nabla F(\hat{\mathbf{w}}) < 0$ for some $\hat{\mathbf{w}}$ where $U_i(\mathbf{w}) = \text{AGG}_i(V_1(\mathbf{w}), \cdots, B_1, \cdots, B_f, \cdots, V_n(\mathbf{w}))$.

Let $L$ denote the Lipschitz parameter of our loss function, *i.e.*, $L = \sup_{\mathbf{w} \in \Omega} \|\nabla F(\mathbf{w})\|$ where $\Omega$ denotes the parameter space. Then we have the following lower bound:

$$-\lambda_i L \leq -\|\mathbb{E}[U_i(\hat{\mathbf{w}})]\|\|\nabla F(\hat{\mathbf{w}})\| \leq \mathbb{E}[U_i(\hat{\mathbf{w}})]^\top \nabla F(\hat{\mathbf{w}}) < 0 \tag{4}$$

where $\lambda_i = \sup_{\mathbf{w} \in \Omega} \|\mathbb{E}[U_i(\mathbf{w})]\|$.

Suppose that an adversary can successfully attack $q$ aggregation rules (W.L.O.G. assume $\text{AGG}_1, \cdots, \text{AGG}_q$ are compromised). We denote

$$\lambda = \max_{i \in [q]} \lambda_i.$$

Let $\mathbf{w} \in \Omega$. Other aggregation rules $\text{AGG}_{m'}$ for $m' \in [M] \setminus [q]$ are resilient against this attack and satisfy

$$\mathbb{E}[U_{m'}(\mathbf{w})]^\top \nabla F(\mathbf{w}) \geq \beta_{m'}$$

for some $\beta_{m'} > 0$.

Note that $\beta_{m'}$ depends on the the gradient variance and bounds on the heterogeneity of gradients across workers. In Section 4, we obtain $\beta$ for a generalized version of Krum.

Recall that MixTailor outputs a rule $U(\mathbf{w}) \in \{U_1(\mathbf{w}), \cdots, U_M(\mathbf{w})\}$ uniformly at random. Combining the lower bounds above, we have

$$\mathbb{E}[U(\mathbf{w})]^\top \nabla F(\mathbf{w}) \geq \frac{M-q}{M}\big(\min_{m'} \beta_{m'}\big) - \frac{q}{M}\lambda L.$$

Finally, a sufficient condition for Byzantine-resilience in term of Definition 1 is that $M$ is large enough to satisfy:

$$\frac{M}{q} > 1 + \frac{\lambda L}{\min_{m'} \beta_{m'}}.$$

### A.4 Optimal and alternative training-time attack design

In this section, we first define an optimal training-time attack.

**Definition 2.** *Let* AGG *denote an aggregation rule. An optimal attack is any vector* $[B_1, B_2, \cdots, B_f]^\top \in \mathbb{R}^{fd}$ *such that*

$$\mathrm{AGG}(B_1, \cdots, B_f, V_{f+1}, \cdots, V_n)^\top \sum_{i=1}^n V_i/n \leq 0.$$

By optimality of an attack, we refer to almost sure convergence guarantees for the problem $\max_{\mathbf{w}} F(\mathbf{w})$ instead of the original problem in (1). Assuming such an attack exists given a pool of aggregators, along the lines of (Bottou, 1998, Section 5.2), the outputs of the aggregation rule are guaranteed to converge to local maxima of $F$, *i.e.,* the attack provably corrupts training. In the following, we consider an alternative attack design based on a min-max problem.

**Alternative attack design.** Now suppose there is a similarity filter, which rejects all similar updates. To circumvent such a filter, the adversary can design a tailored attack by solving the following min max problem:[8]

$$\mathcal{P}_1: \min_{(B_1, \cdots, B_f)} \max_{m \in [M]} \mathrm{AGG}_m(B_1, \cdots, B_f, V_{f+1}, \cdots, V_n)^\top \sum_{i=1}^n \frac{V_i}{n}$$

$$\text{subject to } \|B_i - B_j\| \geq \epsilon, \quad \forall (i,j).$$

Let $\xi \triangleq \max_{m \in [M]} \mathrm{AGG}_m(B_1, \cdots, B_f, V_{f+1}, \cdots, V_n)^\top$
$\sum_{i=1}^n V_i/n$ denote the solution to the inner maximization problem. Then $\mathcal{P}_1$ is equivalent to the following problem:

$$\mathcal{P}_2: \min_{(B_1, \cdots, B_f), \xi} \xi$$

$$\text{subject to } \mathrm{AGG}_m(B_1, \cdots, B_f, V_{f+1}, \cdots, V_n)^\top \sum_{i=1}^n \frac{V_i}{n} \leq \xi, \; \forall m,$$

$$\|B_i - B_j\| \geq \epsilon \quad \forall (i,j).$$

We note that if the optimal value of $\mathcal{P}_2$ is negative, *i.e.,* $\xi(\epsilon)^* < 0$, the solution will be a theoretically guaranteed and successful attack to circumvent MixTailor algorithm. Note that such an attack circumvents the similarity filter too. $\mathcal{P}_2$ can be infeasible. In general, $\mathcal{P}_2$ is an NP-hard problem due to complex and nonconvex constraints. Even by ignoring a similarity filter and restricting the search space, the adversary should solve this nonconvex problem:

$$\mathcal{P}_3: \min_{\lambda, \xi} \xi$$

---

[8]Suboptimal attacks, attacks under partial knowledge, and model-based tailored attacks can be designed along the lines of Appendix A.1.

$$\text{subject to } \mathbf{v}_m(\lambda)^\top \sum_{i=1}^n \frac{V_i}{n} \le \xi, \quad \forall m \in [M].$$

where $\mathbf{v}_m(\lambda) = \mathrm{AGG}_m(\underbrace{-\lambda \sum_{i=1}^n V_i, \cdots, -\lambda \sum_{i=1}^n V_i}_{f}, V_{f+1}, \cdots, V_n).$

We note the $\mathcal{P}_3$ is not guaranteed to be feasible. If feasible, a solution might be obtained by relaxing the nonconvex constraint to a convex one. Developing efficient algorithms to solve this problem approximately is an interesting future direction.

**Intersection of upper bounds in (Fang et al., 2020, Theorem 1) and lower bounds in (Xie et al., 2020b, Theorem 1).** In (Fang et al., 2020, Theorem 1), an upper bound is established on the norm of the attack vector that is tailored against Krum. In particular, $\lambda = \mathcal{O}(1/\sqrt{d})$ fails Krum. Let $\overline{\mathbf{g}} = \frac{1}{n-f}\sum_{i=f+1}^n \mathbb{E}[\mathbf{g}_i]$ denote expected value of honest updates sent by good workers. Building on a similar argument as in (Xie et al., 2020b, Theorem 1) and (Hawkins, 1971, Theorem 1(b)), a lower bound on $\lambda$ tailored against comed is given by $\lambda = \Omega\left(\left|1 - \frac{\hat{\sigma}}{\sqrt{n-f-1}\|\overline{\mathbf{g}}\|_\infty}\right|\right)$ where $\hat{\sigma}$ is the coordinate-wise variance defined in (Xie et al., 2020b, Theorem 1). A sufficient condition that guarantees emptiness of the intersection for an attack, which fails both Krum and comed is that the variance $\hat{\sigma}$ is large enough such that $\frac{\hat{\sigma}}{\sqrt{n-f-1}\|\overline{\mathbf{g}}\|_\infty} \ge 1 - \frac{1}{\sqrt{d}}.$

### A.5 Proof of Theorem 1

Let $i^*$ denote the index of the worker selected by (3). Using Jensen's inequality, we have

$$\|\mathbb{E}[U] - \nabla F\|_2^2 = \left\|\mathbb{E}\left[U - \frac{1}{|\mathcal{N}_g(i^*)|}\sum_{j\in\mathcal{N}_g(i^*)} V_j\right]\right\|_2^2$$
$$\le \mathbb{E}\left[\left\|U - \frac{1}{|\mathcal{N}_g(i^*)|}\sum_{j\in\mathcal{N}_g(i^*)} V_j\right\|_2^2\right].$$

The law of total expectation implies

$$\|\mathbb{E}[U] - \nabla F\|_2^2 \le \sum_{i=1}^{n-f} \mathbb{E}\left[\left\|V_i - \frac{1}{|\mathcal{N}_g(i)|}\sum_{j\in\mathcal{N}_g(i)} V_j\right\|_2^2\right]\Pr(i^* = i)$$
$$+ \sum_{k=1}^{f} \mathbb{E}\left[\left\|B_k - \frac{1}{|\mathcal{N}_g(k)|}\sum_{j\in\mathcal{N}_g(k)} V_j\right\|_2^2\right]\Pr(i^* = k).$$

In the following we develop upper bounds that hold for each conditional expectation uniformly over $i \in [n-f]$ and $k \in [f]$.

Let $V_i \in \mathcal{G}$. We first find an upper bound on $\mathbb{E}\left[\left\|V_i - \frac{1}{|\mathcal{N}_g(i)|}\sum_{j\in\mathcal{N}_g(i)} V_j\right\|_2^2\right]$. Using the Jensen's inequity and the variance upper bound $\sigma^2$, we can show that

$$\mathbb{E}\left[\left\|V_i - \frac{1}{|\mathcal{N}_g(i)|}\sum_{j\in\mathcal{N}_g(i)} V_j\right\|_2^2\right] \le \frac{1}{|\mathcal{N}_g(i)|}\sum_{j\in\mathcal{N}_g(i)} \mathbb{E}\left[\left\|V_i - V_j\right\|_2^2\right]$$
$$\le 2\sigma^2.$$

Note that above upper bound holds for all $V_i \in \mathcal{G}$. Now let $B_k \in \mathcal{B}$ denote a Byzantine worker that is selected by (3). For all good $V_i \in \mathcal{G}$, we have

$$\sum_{j \in \mathcal{N}_g(k)} \left\| B_k - V_j \right\|_p^2 + \sum_{l \in \mathcal{N}_b(k)} \left\| B_k - B_l \right\|_p^2$$
$$\leq \sum_{j \in \mathcal{N}_g(i)} \left\| V_i - V_j \right\|_p^2 + \sum_{l \in \mathcal{N}_b(i)} \left\| V_i - B_l \right\|_p^2. \tag{5}$$

Note that Jensen's inequality implies that

$$\mathbb{E}\left[ \left\| B_k - \frac{1}{|\mathcal{N}_g(k)|} \sum_{j \in \mathcal{N}_g(k)} V_j \right\|_2^2 \right] \leq \frac{1}{|\mathcal{N}_g(k)|} \sum_{j \in \mathcal{N}_g(k)} \mathbb{E}\left[ \left\| B_k - V_j \right\|_2^2 \right].$$

In the rest of our proof, we use the following lemma.

**Lemma 1.** *Let $\mathbf{x} \in \mathbb{R}^d$. Then, for all $0 < p < q$, we have $\|\mathbf{x}\|_q \leq \|\mathbf{x}\|_p \leq d^{1/p-1/q}\|\mathbf{x}\|_q$.*

Let $p \geq 2$. By Lemma 1, we have

$$\|B_k - V_j\|_2^2 \leq d^{1-2/p}\|B_k - V_j\|_p^2.$$

Combining the above inequality with (5), it follows that

$$\frac{1}{|\mathcal{N}_g(k)|} \sum_{j \in \mathcal{N}_g(k)} \mathbb{E}\left[ \left\| B_k - V_j \right\|_2^2 \right] \leq \frac{d^{1-2/p}}{|\mathcal{N}_g(k)|} \left( \sum_{j \in \mathcal{N}_g(i)} \mathbb{E}\left[ \left\| V_i - V_j \right\|_p^2 \right] \right.$$
$$\left. + \sum_{l \in \mathcal{N}_b(i)} \mathbb{E}\left[ \left\| V_i - B_l \right\|_p^2 \right] \right). \tag{6}$$

The following lemma is useful for our proofs.

**Lemma 2.** *Let $i \in [n]$. Then we have*

$$0 \leq \mathcal{N}_b(i) \leq f$$
$$n - 2f - 2 \leq \mathcal{N}_g(i) \leq n - f - 2.$$

By Lemmas 1 and 2, we have

$$\frac{d^{1-2/p}}{|\mathcal{N}_g(k)|} \sum_{j \in \mathcal{N}_g(i)} \mathbb{E}\left[ \left\| V_i - V_j \right\|_p^2 \right] \leq \frac{d^{1-2/p}}{|\mathcal{N}_g(k)|} \sum_{j \in \mathcal{N}_g(i)} \mathbb{E}\left[ \left\| V_i - V_j \right\|_2^2 \right]$$
$$\leq 2\sigma^2 d^{1-2/p} \frac{|\mathcal{N}_g(i)|}{|\mathcal{N}_g(k)|}$$
$$\leq 2\sigma^2 d^{1-2/p} \frac{n - f - 2}{n - 2f - 2}.$$

Since $f < n/2$, for each $l \in \mathcal{N}_b(i)$ and $V_i \in \mathcal{G}$, there exists an $\zeta(i)$ such that $V_{\zeta(i)} \in \mathcal{G}$ such that

$$\|V_i - B_l\|_p^2 \leq \|V_i - V_{\zeta(i)}\|_p^2.$$

By Lemma 1 and the law to total expectation, for all $V_i \in \mathcal{G}$, we have

$$\mathbb{E}[\|V_i - V_{\zeta(i)}\|_p^2] \leq \mathbb{E}[\|V_i - V_{\zeta(i)}\|_2^2]$$
$$\leq \mathbb{E}[\|V_i - V_j\|_2^2]\Pr(\zeta(i) = j)$$
$$\leq 2\sigma^2.$$

By Lemma 2, we have

$$\frac{d^{1-2/p}}{|\mathcal{N}_g(k)|}\sum_{l \in \mathcal{N}_b(i)} \mathbb{E}\left[\left\|V_i - B_l\right\|_p^2\right] \leq 2d^{1-2/p}\sigma^2\frac{|\mathcal{N}_b(i)|}{|\mathcal{N}_g(k)|}$$
$$\leq 2d^{1-2/p}\sigma^2\frac{f}{n-2f-2}.$$

Combining above upper bounds, we have $\|\mathbb{E}[U(\mathbf{w})] - \nabla F(\mathbf{w})\|_2^2 \leq 2\sigma^2\left(1 + d^{1-2/p}\left(\frac{n-f-2}{n-2f-2} + \frac{f}{n-2f-2}\right)\right)$.

Now let $1 \leq p \leq 2$. By Lemma 1, we have

$$\xi(\mathcal{N}_g(k)) \leq \sum_{j \in \mathcal{N}_g(k)} \left\|B_k - V_j\right\|_p^2$$
$$\leq \sum_{j \in \mathcal{N}_g(i)} \left\|V_i - V_j\right\|_p^2 + \sum_{l \in \mathcal{N}_b(i)} \left\|V_i - B_l\right\|_p^2$$
$$\leq d^{2/p-1}\left(\sum_{j \in \mathcal{N}_g(i)} \left\|V_i - V_j\right\|_2^2 + \sum_{l \in \mathcal{N}_b(i)} \left\|V_i - B_l\right\|_2^2\right).$$

where $\xi(\mathcal{N}_g(k)) = \sum_{j \in \mathcal{N}_g(k)} \left\|B_k - V_j\right\|_2^2$.

Following a similar approach, we obtain $\|\mathbb{E}[U(\mathbf{w})] - \nabla F(\mathbf{w})\|_2^2 \leq 2\sigma^2\left(1 + d^{2/p-1}\left(\frac{n-f-2}{n-2f-2} + \frac{f}{n-2f-2}\right)\right)$.

For the last part of the proof, using the law of total expectation, we have

$$\mathbb{E}[\|U\|_2^r] \leq \mathbb{E}[\|G\|_2^r] + \sum_{k=1}^{f} \mathbb{E}[\|B_k\|_2^r]\Pr(i^* = k).$$

Let $B_k \in \mathcal{B}$ denote a Byzantine worker that is selected by (3). Using Lemma 1, for all good $V_i \in \mathcal{G}$, we have

$$\left\|B_k - \frac{1}{|\mathcal{N}_g(k)|}\sum_{j \in \mathcal{N}_g(k)} V_j\right\|_2 \leq \sqrt{\Delta_p}$$
$$\leq C\sum_{i=1}^{n-f} \|V_i\|_2$$

where $\Delta_p = \frac{C_{d,p}}{|\mathcal{N}_g(k)|}\sum_{j \in \mathcal{N}_g(i)} \|V_i - V_j\|_p^2 + \frac{C_{d,p}|\mathcal{N}_b(i)|}{|\mathcal{N}_g(k)|}\|V_i - V_{\zeta(i)}\|_p^2$ and $C_{d,p} = d^{\frac{\max\{p,2\}-\min\{p,2\}}{p}}$. Note that

$$\|B_k\|_2 \leq \left\|B_k - \frac{1}{|\mathcal{N}_g(k)|}\sum_{j \in \mathcal{N}_g(k)} V_j\right\|_2 + \left\|\frac{1}{|\mathcal{N}_g(k)|}\sum_{j \in \mathcal{N}_g(k)} V_j\right\|_2$$
$$\leq C\sum_{i=1}^{n-f} \|V_i\|_2.$$

This follows that

$$\|B_k\|_2^r \leq C\sum_{r_1+\cdots+r_{n-f}=r} \|V_1\|_2^{r_1}\cdots\|V_{n-f}\|_2^{r_{n-f}}.$$

Taking expectation and applying weighted AM–GM inequality, we have

$$\mathbb{E}[\|B_k\|_2^r] \leq C \sum_{r_1 + \cdots + r_{n-f} = r} \left( \frac{r_1}{r} \|V_1\|_2^{r_1} + \cdots + \frac{r_{n-f}}{r} \|V_{n-f}\|_2^{r_{n-f}} \right)$$
$$\leq C\mathbb{E}[\|G\|_2^r].$$

This completes the proof.

### A.6 Proof of Theorem 2

Let $i^*$ denote the index of the worker selected by (3). Using Jensen's inequality, we have

$$\|\mathbb{E}[U] - \nabla F\|_2^2 = \left\| \mathbb{E}\left[ U - \frac{1}{n-f} \sum_{j=1}^{n-f} V_j \right] \right\|_2^2$$
$$\leq \mathbb{E}\left[ \left\| U - \frac{1}{n-f} \sum_{j=1}^{n-f} V_j \right\|_2^2 \right].$$

The law of total expectation implies

$$\|\mathbb{E}[U] - \nabla F\|_2^2 \leq \sum_{i=1}^{n-f} \mathbb{E}\left[ \left\| V_i - \frac{1}{n-f} \sum_{j=1}^{n-f} V_j \right\|_2^2 \right] \Pr(i^* = i)$$
$$+ \sum_{k=1}^{f} \mathbb{E}\left[ \left\| B_k - \frac{1}{n-f} \sum_{j=1}^{n-f} V_j \right\|_2^2 \right] \Pr(i^* = k).$$

In the following we develop upper bounds that hold for each conditional expectation uniformly over $i \in [n-f]$ and $k \in [f]$.

Let $V_i \in \mathcal{G}$. We first find an upper bound on $\mathbb{E}\left[ \left\| V_i - \frac{1}{n-f} \sum_{j=1}^{n-f} V_j \right\|_2^2 \right]$. We use the following lemma in our proofs.

**Lemma 3.** *Let* $\mathbf{u}, \mathbf{v} \in \mathbb{R}^d$. *Then we have*

$$\|\mathbf{u} + \mathbf{v}\|^2 \leq 2\|\mathbf{u}\|^2 + 2\|\mathbf{v}\|^2.$$

Using the Jensen's inequity and Lemma 3 under Assumption 1, we can show that

$$\delta(V_i) \leq \frac{1}{n-f} \sum_{j=1}^{n-f} \mathbb{E}\left[ \left\| V_i - V_j \right\|_2^2 \right]$$
$$\leq \mathbb{E}\left[ \left\| V_i - \mathbf{g}_i \right\|_2^2 \right] + \frac{1}{n-f} \sum_{j=1}^{n-f} \mathbb{E}\left[ \left\| V_j - \mathbf{g}_i \right\|_2^2 \right]$$
$$\leq 2\sigma^2 + \frac{1}{n-f} \sum_{j=1}^{n-f} \|\mathbf{g}_j - \mathbf{g}_i\|_2^2$$
$$\leq 2\sigma^2 + 2\|\mathbf{g}_i - \nabla F\|_2^2 + \frac{2}{n-f} \sum_{j=1}^{n-f} \|\mathbf{g}_j - \nabla F\|_2^2$$
$$\leq 2\sigma^2 + 2(n-f+1)\Delta^2$$

where $\delta(V_i) = \mathbb{E}\left[ \left\| V_i - \frac{1}{n-f} \sum_{j=1}^{n-f} V_j \right\|_2^2 \right]$.

Note that above upper bound holds for all $V_i \in \mathcal{G}$. Now let $B_k \in \mathcal{B}$ denote a Byzantine worker that is selected by (3). By Lemma 3, we have

$$\mathbb{E}\left[\left\|B_k - \frac{1}{n-f}\sum_{j=1}^{n-f}V_j\right\|_2^2\right] \le 2\mathbb{E}\left[\left\|B_k - \frac{1}{|\mathcal{N}_g(k)|}\sum_{j\in\mathcal{N}_g(k)}V_j\right\|_2^2\right]$$

$$+ 2\mathbb{E}\left[\left\|\frac{1}{|\mathcal{N}_g(k)|}\sum_{j\in\mathcal{N}_g(k)}V_j - \frac{1}{n-f}\sum_{l=1}^{n-f}V_l\right\|_2^2\right].$$

We first find an upper bound on the second term. Note that Jensen's inequality implies that

$$\delta(\mathcal{N}_g(k)) \le \frac{1}{|\mathcal{N}_g(k)|}\sum_{j\in\mathcal{N}_g(k)}\mathbb{E}\left[\left\|V_j - \frac{1}{n-f}\sum_{l=1}^{n-f}V_l\right\|_2^2\right]$$

$$\le \frac{1}{|\mathcal{N}_g(k)|}\sum_{j\in\mathcal{N}_g(k)}\left(\frac{1}{n-f}\sum_{l=1}^{n-f}\mathbb{E}\left[\left\|V_j - V_l\right\|_2^2\right]\right)$$

$$\le 2\sigma^2 + \frac{1}{|\mathcal{N}_g(k)|}\sum_{j\in\mathcal{N}_g(k)}\left(\frac{1}{n-f}\sum_{l=1}^{n-f}\|\mathbf{g}_j - \mathbf{g}_l\|_2^2\right)$$

$$\le 2\sigma^2 + 2\left(\frac{n-f}{|\mathcal{N}_g(k)|} + 1\right)\Delta^2$$

$$\le 2\sigma^2 + 2\left(\frac{n-f}{n-2f-2} + 1\right)\Delta^2$$

where $\delta(\mathcal{N}_g(k)) = \mathbb{E}\left[\left\|\frac{1}{|\mathcal{N}_g(k)|}\sum_{j\in\mathcal{N}_g(k)}V_j - \frac{1}{n-f}\sum_{l=1}^{n-f}V_l\right\|_2^2\right]$.

We now find an upper bound on $\mathbb{E}\left[\left\|B_k - \frac{1}{|\mathcal{N}_g(k)|}\sum_{j\in\mathcal{N}_g(k)}V_j\right\|_2^2\right]$. The Jensen's inequality implies that $\mathbb{E}\left[\left\|B_k - \frac{1}{|\mathcal{N}_g(k)|}\sum_{j\in\mathcal{N}_g(k)}V_j\right\|_2^2\right] \le \frac{1}{|\mathcal{N}_g(k)|}\sum_{j\in\mathcal{N}_g(k)}\mathbb{E}\left[\left\|B_k - V_j\right\|_2^2\right]$. For all good $V_i \in \mathcal{G}$, we have

$$\sum_{j\in\mathcal{N}_g(k)}\left\|B_k - V_j\right\|_p^2 + \sum_{l\in\mathcal{N}_b(k)}\left\|B_k - B_l\right\|_p^2$$

$$\le \sum_{j\in\mathcal{N}_g(i)}\left\|V_i - V_j\right\|_p^2 + \sum_{l\in\mathcal{N}_b(i)}\left\|V_i - B_l\right\|_p^2. \tag{7}$$

Let $p \ge 2$. By Lemma 1 and inequality (7), we have

$$\frac{1}{|\mathcal{N}_g(k)|}\sum_{j\in\mathcal{N}_g(k)}\mathbb{E}\left[\left\|B_k - V_j\right\|_2^2\right] \le \frac{d^{1-2/p}}{|\mathcal{N}_g(k)|}\left(\sum_{j\in\mathcal{N}_g(i)}\mathbb{E}\left[\left\|V_i - V_j\right\|_p^2\right]\right.$$

$$\left. + \sum_{l\in\mathcal{N}_b(i)}\mathbb{E}\left[\left\|V_i - B_l\right\|_p^2\right]\right). \tag{8}$$

By Lemmas 1 and 2, we have

$$\tilde{\Delta}_p \le \frac{d^{1-2/p}}{|\mathcal{N}_g(k)|}\sum_{j\in\mathcal{N}_g(i)}\mathbb{E}\left[\left\|V_i - V_j\right\|_2^2\right]$$

$$\le d^{1-2/p}\frac{1}{|\mathcal{N}_g(k)|}\left(2\sigma^2|\mathcal{N}_g(i)| + \sum_{j\in\mathcal{N}_g(i)}\|\mathbf{g}_i - \mathbf{g}_j\|_2^2\right)$$

$$\le \left(2\sigma^2 + 4(n-f)\Delta^2\right)d^{1-2/p}\frac{|\mathcal{N}_g(i)|}{|\mathcal{N}_g(k)|}$$

$$\leq \left(2\sigma^2 + 4(n-f)\Delta^2\right)d^{1-2/p}\frac{n-f-2}{n-2f-2}$$

where $\tilde{\Delta}_p = \frac{d^{1-2/p}}{|\mathcal{N}_g(k)|}\sum_{j\in\mathcal{N}_g(i)}\mathbb{E}\left[\left\|V_i - V_j\right\|_p^2\right]$.

Since $f < n/2$, for each $l \in \mathcal{N}_b(i)$ and $V_i \in \mathcal{G}$, there exists an $\zeta(i)$ such that $V_{\zeta(i)} \in \mathcal{G}$ such that

$$\|V_i - B_l\|_p^2 \leq \|V_i - V_{\zeta(i)}\|_p^2.$$

By Lemma 1 and the law to total expectation, for all $V_i \in \mathcal{G}$, we have

$$\begin{aligned}
\mathbb{E}[\|V_i - V_{\zeta(i)}\|_p^2] &\leq \mathbb{E}[\|V_i - V_{\zeta(i)}\|_2^2] \\
&\leq \mathbb{E}[\|V_i - V_j\|_2^2]\Pr(\zeta(i) = j) \\
&\leq 2\sigma^2 + 4(n-f)\Delta^2.
\end{aligned}$$

By Lemma 2, we have

$$\begin{aligned}
\hat{\Delta}_p &\leq \left(2\sigma^2 + 4(n-f)\Delta^2\right)d^{1-2/p}\frac{|\mathcal{N}_b(i)|}{|\mathcal{N}_g(k)|} \\
&\leq \left(2\sigma^2 + 4(n-f)\Delta^2\right)d^{1-2/p}\frac{f}{n-2f-2}
\end{aligned}$$

where $\hat{\Delta}_p = \frac{d^{1-2/p}}{|\mathcal{N}_g(k)|}\sum_{l\in\mathcal{N}_b(i)}\mathbb{E}\left[\left\|V_i - B_l\right\|_p^2\right]$.

Combining above upper bounds, we have $\|\mathbb{E}[U(\mathbf{w})] - \nabla F(\mathbf{w})\|_2^2 \leq C_1 + C_2 d^{1-2/p}\left(\frac{n-f-2}{n-2f-2} + \frac{f}{n-2f-2}\right)$.

Now let $1 \leq p \leq 2$. By Lemma 1, we have

$$\begin{aligned}
\xi(\mathcal{N}_g(k)) &\leq \sum_{j\in\mathcal{N}_g(k)}\left\|B_k - V_j\right\|_p^2 \\
&\leq \sum_{j\in\mathcal{N}_g(i)}\left\|V_i - V_j\right\|_p^2 + \sum_{l\in\mathcal{N}_b(i)}\left\|V_i - B_l\right\|_p^2 \\
&\leq d^{2/p-1}\left(\sum_{j\in\mathcal{N}_g(i)}\left\|V_i - V_j\right\|_2^2 + \sum_{l\in\mathcal{N}_b(i)}\left\|V_i - B_l\right\|_2^2\right)
\end{aligned}$$

where $\xi(\mathcal{N}_g(k)) = \sum_{j\in\mathcal{N}_g(k)}\left\|B_k - V_j\right\|_2^2$.

Following a similar approach, we obtain $\|\mathbb{E}[U(\mathbf{w})] - \nabla F(\mathbf{w})\|_2^2 \leq C_1 + C_2 d^{2/p-1}\left(\frac{n-f-2}{n-2f-2} + \frac{f}{n-2f-2}\right)$.

Finally, under Assumption 2 and using the law of total expectation, we have

$$\mathbb{E}[\|U\|_2^r] \leq \sum_{i=1}^{n-f}K_{r,i}\mathbb{E}[\|G\|_2^r] + \sum_{k=1}^{f}\mathbb{E}[\|B_k\|_2^r]\Pr(i^* = k).$$

Let $B_k \in \mathcal{B}$ denote a Byzantine worker that is selected by (3). Using Lemma 1, for all good $V_i \in \mathcal{G}$, we have

$$\begin{aligned}
\left\|B_k - \frac{1}{|\mathcal{N}_g(k)|}\sum_{j\in\mathcal{N}_g(k)}V_j\right\|_2 &\leq \sqrt{\Delta_p} \\
&\leq C\sum_{i=1}^{n-f}\|V_i\|_2
\end{aligned}$$

**Table 2:** Training hyper-parameters for CIFAR-10 and MNIST. The network architecture is a 4 layer neural net with 2 convolutional layers and two fully connected layers. Drop out is used between the convulutional layers and the fully connected layers.

| Hyper-parameter | CIFAR-10 | MNIST |
|---|---|---|
| Learning Rate | 0.001 | 0.001 |
| Batch Size | 80 | 50 |
| Momentum | 0.9 | 0.9 |
| Total Iterations | 40K | 50K |
| Weight Decay | $10^{-4}$ | $10^{-4}$ |

and $\Delta_p = \frac{C_{d,p}}{|\mathcal{N}_g(k)|} \sum_{j \in \mathcal{N}_g(i)} \|V_i - V_j\|_p^2 + \frac{C_{d,p}|\mathcal{N}_b(i)|}{|\mathcal{N}_g(k)|} \|V_i - V_{\zeta(i)}\|_p^2$ and $C_{d,p} = d^{\frac{\max\{p,2\} - \min\{p,2\}}{p}}$. Furthermore, we have

$$\|B_k\|_2 \leq \left\|B_k - \frac{1}{|\mathcal{N}_g(k)|} \sum_{j \in \mathcal{N}_g(k)} V_j\right\|_2 + \left\|\frac{1}{|\mathcal{N}_g(k)|} \sum_{j \in \mathcal{N}_g(k)} V_j\right\|_2$$

$$\leq C \sum_{i=1}^{n-f} \|V_i\|_2.$$

This follows that

$$\|B_k\|_2^r \leq C \sum_{r_1 + \cdots + r_{n-f} = r} \|V_1\|_2^{r_1} \cdots \|V_{n-f}\|_2^{r_{n-f}}.$$

Taking expectation and applying weighted AMâĂŞGM inequality, we have $\mathbb{E}[\|B_k\|_2^r] \leq C \sum_{r_1 + \cdots + r_{n-f} = r} \left(\frac{r_1}{r} \|V_1\|_2^{r_1} + \cdots + \frac{r_{n-f}}{r} \|V_{n-f}\|_2^{r_{n-f}}\right) \leq C\mathbb{E}[\|G\|_2^r]$, which completes the proof.

### A.7  Proof of Theorem 3

Let $\mathcal{R} = \{\mathbf{w} | \|\mathbf{w}\|_2^2 \geq R\}$ denote a horizon. Following Theorem 2, we note that

$$2\mathbb{E}[U(\mathbf{w})]^\top \nabla F(\mathbf{w}) \geq \|\mathbb{E}[U(\mathbf{w})]\|_2^2 + \|\nabla F(\mathbf{w})\|_2^2 - \tilde{C}$$

$$\geq \inf_{\mathbf{w} \in \mathcal{R}} \left(\|\mathbb{E}[U(\mathbf{w})]\|_2^2 + \|\nabla F(\mathbf{w})\|_2^2\right) - \tilde{C}$$

$$\geq \inf_{\mathbf{w} \in \mathcal{R}} \|\mathbb{E}[U(\mathbf{w})]\|_2^2 + \inf_{\mathbf{w} \in \mathcal{R}} \|\nabla F(\mathbf{w})\|_2^2 - \tilde{C}$$

$$\geq \beta$$

where $\tilde{C} = C_1 + C_2 \Lambda(n, f, d, p)$.

Combining this bound with

$$\inf_{\mathbf{w} \in \mathcal{R}} \frac{\mathbf{w}^\top \nabla F(\mathbf{w})}{\|\mathbf{w}\|_2 \|\nabla F(\mathbf{w})\|_2} \geq \cos(\theta)$$

and noting $\theta + \sup_{\mathbf{w} \in \mathcal{R}} \alpha < \pi/2$, we have $\mathbf{w}^\top \mathbb{E}[U(\mathbf{w})] > 0$ for $\mathbf{w} \in \mathcal{R}$. The rest of the proof follows along the lines of (Fisk, 1965; Métivier, 1982; Bottou, 1998).

We note that median-based aggregators such as Krum, comed, and Bulyan *do not necessarily output an unbiased estimate of the gradient of the true empirical risk* (Karimireddy et al., 2021, Section 3).

### A.8  Experimental details and additional experiments

In this section, we run a series of experiments to find out the effect of individual aggregators in the pool of aggregators. As explained in the main body, the pool of aggregators contains four different aggregation mechanism. In this section, we remove one aggreagtor from the pool, to see what will be the effect.

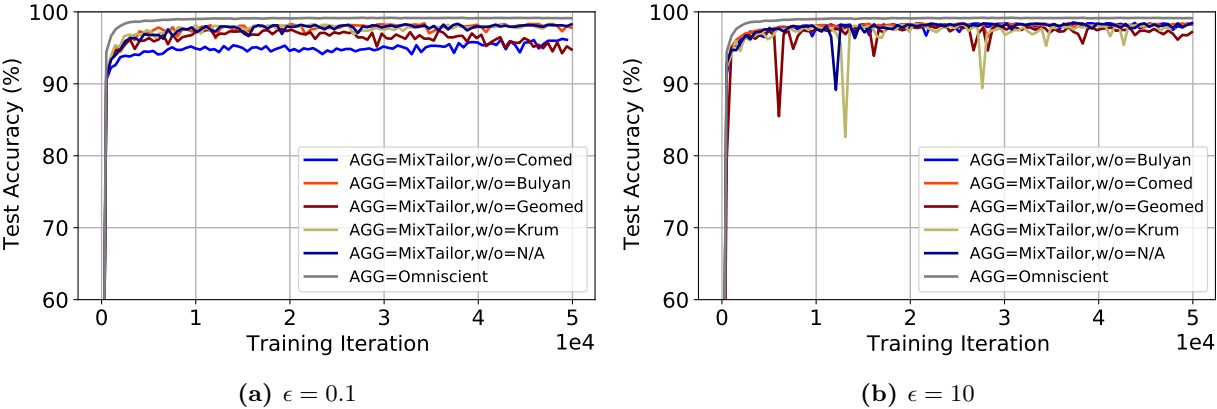

**(a)** $\epsilon = 0.1$ **(b)** $\epsilon = 10$

**Figure 5:** *Test accuracy on MNIST for the tailored attack, which is applied at each iteration. The total number of workers and the number of Byzantines are set to $n = 12$ and $f = 2$, respectively. The dataset is randomly and equally partitioned among workers. The omniscient aggregator receives and averages 10 honest gradients at each iteration.*

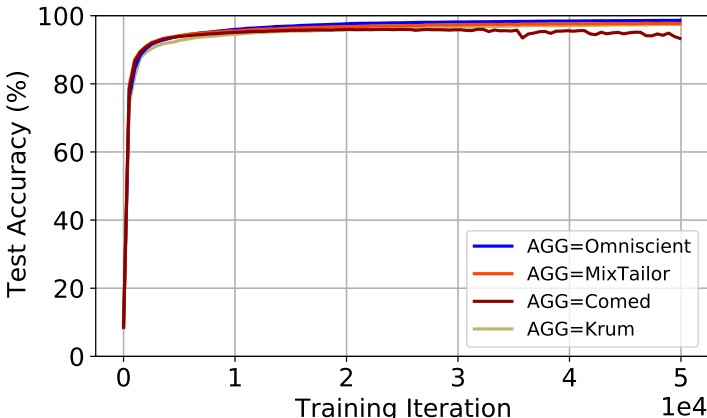

**Figure 6:** *Test accuracy on MNIST under "A Little" attack (Baruch et al., 2019). The total number of workers and the number of Byzantines are set to $n = 12$ and $f = 2$, respectively. The dataset is randomly and equally partitioned among workers. The omniscient aggregator receives and averages 10 honest gradients at each iteration.*

**Details of implementation.** The details of training hyper-parameters are shown in Table 2. The network architecture is a 4 layer neural net with 2 convolutional layers + two fully connected layers. Drop out is used between the convulutional layers and the fully connected layers. All experiments where run on single-GPU machines using a cluster that had access to T4, RTX6000, and P100 GPUs.

Fig. 5 shows the result when one of the aggregators is removed from the pool. The w/o tag represents the aggregator that is not included in the pool. We observe that under both attacks, removing the Bulyan from the pool of aggregators increases the validation accuracy by around 2%. Also, it is worth noting that removing the geometric median reduces the accuracy, which is expected since these tailored attacks are designed for Krum and comed.

Fig. 6 shows the performance of MixTailor under "A Little" attack in (Baruch et al., 2019). To observe resilience of different aggregation methods decoupled from momentum, we set momentum to zero in this experiment. We observe that all aggregation methods including MixTailor are resilient against "A Little" attack in this setting. This observation confirms that tailored attacks in (Fang et al., 2020; Xie et al., 2020b) are more effective than "A Little" attack. We emphasize that both Krum and comed fail under carefully designed tailored attacks with $f = 2$ Byzantines with and without momentum, which is not the case for "A Little" attack.

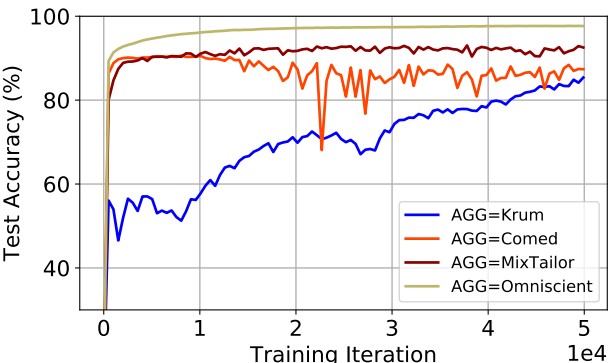

**Figure 7:** *Test accuracy on MNIST under adaptive attack. MixTailor is robust to the adaptive attack. The rest of the setup is similar to Fig. 1.*

**MixTailor under an adaptive attack.** We have considered a stronger and adaptive attacker which optimizes its attack by enumerating over a set of $\epsilon$'s and selects *the worst $\epsilon$ against the aggregator at every single iteration.* The adversary enumerates among all those $\epsilon$'s and finds out which one is the most effective attack by applying the aggregator (the attacker simulates the server job by applying the aggregator with different $\epsilon$'s and finds the best attack and then outputs the best attack for the server to aggregate). Regarding MixTailor, the attacker selects a random aggregator from the MixTailor's aggregator pool in each iteration and finds the worst epsilon corresponding to this aggregator. The attacker finds *an adaptive* attack by calculating the dot product of the output of the aggregator and the direction of aggregated gradients without attacks when different epsilons are fed into the aggregator. The attacker chooses the epsilon that causes the aggregator to produce the gradient that has the smallest dot product with the true gradient. In Fig. 7, we ran this experiment over MNIST and observe that MixTailor is able to outperform both Krum and Comed. Comed's accuracy changes between 85-87%, Krum's accuracy is 83-85%, and MixTailor's accuracy is 91.80-92.55%. The accuracy of the omniscient aggregator is 97.62-97.68%. The set of epsilons used by the adaptive attacker is 0.1, 0.5, 1, and 10.

