# OpenReview forum: "MixTailor: Mixed Gradient Aggregation for Robust Learning Against Tailored Attacks"
_TMLR — Accepted by TMLR_

### Review · Reviewer_mqSq · 2022-07-27

**Summary Of Contributions:**

The paper propose to randomize the robust aggregation process in federated learning to defend against customized attacks on the robust aggregators. For both iid and non-iid settings, the authors establish almost sure convergence guarantees.

**Requested Changes:**

1. Please compare with sub-sampling.
2. Please properly cite the relevant works and evaluate how they compare with mixtailor

**Strengths And Weaknesses:**

I did not check the correctness of the convergence theorem but I believe that the claims made in the submission are supported by the empirical evaluation.

Strength:
1. The idea to randomize the robust aggregator is novel.
2. The paper is well written and easy to follow.

Weakness:
1. Although the idea of randomizing the robust aggregator is novel, it seems that there is no intrinsic difference between mixtailor and other randomization in federated learning such as sub-sampling. Since sub-sampling can be somewhat bypassed, can the authors discuss why the proposed method is not bypassable?
2. The empirical evaluation only compares with 2 prior works. More robust aggregators should be included.
3. Many recent advances in robust aggregation are ignored including
[1] Byzantine-Resilient Non-Convex Stochastic Gradient Descent
[2] Variance Reduction is an Antidote to Byzantines: Better Rates, Weaker Assumptions and Communication Compression as a Cherry on the Top
[3] Byzantine-Resilient High-Dimensional SGD with Local Iterations on Heterogeneous Data
[4] Byzantine-Robust Federated Learning with Optimal Rates and Privacy Guarantee

---

> ### Author Response · Authors · 2022-09-01
> **Response to Reviewer mqSq Part 1**
>
> We thank the reviewer for their thoughtful comments which we address one by one below.
>
> ------
>
> - Clarify difference between MixTailor and other randomization in federated learning such as sub-sampling. Compare with sub-sampling:
>
>
> Unlike hyperparameter-based randomization techniques such as sub-sampling, MixTailor provides randomness in the *structure of aggregation rules*, which makes it impossible for the attacker to *control the optimization trajectory*. Hyperparameter-based randomization techniques as sub-sampling can also improve robustness by some extent, however, the adversary can still design a successful attack by focusing *on the specific aggregation structure*. The adversary can do so for example by mimicking the subsampling procedure to fail it.
>
> Formally, Suppose that the $M$ aggregators used by the server include $\{\mathrm{AGG}_1, \mathrm{AGG}_2, \ldots, \mathrm{AGG}_M\}.$
>
>
> We note that each aggregation rule $\mathrm{AGG}_i$ is either deterministic or has some hyperparameters which can be set randomly such as sub-sampling parameters. For each $\mathrm{AGG}_i$, we define attack complexity as follows:
>
> Let $T_i(n,f,d,\epsilon)$ denote the *number of elementary operations* an informed adversary requires to design a tailored attack in terms of solving the optimization problem in Section 2.3 with precision $\epsilon$ for satisfying constraints such that $\epsilon=\arg\cos\Big(\frac{-{\bf g}^{\top}{\bf g}’}{\|{\bf g}’\|\|{\bf g}\|}\Big)=\pi-\arg\cos\Big(\frac{{\bf g}^{\top}{\bf g}’}{\|{\bf g}’\|\|{\bf g}\|}\Big)$. For a given aggregation rule, the number of elementary operations increases to achieve smaller values of $\epsilon$, which amounts to optimizing more effective attacks. The attack complexity for MixTailor is ${\Omega}(\sum_{i=1}^M T_i\big(n,f,d,\epsilon)\big)$, which monotonically increases by $M$. To see this, assume there exists an attacker with lower complexity, then the attacker fails to break at least one of the aggregators. Note that precise expressions of $T_i$’s, i.e., the exact numbers of elementary operations depend on the specific problem to solve (dataset, loss, architecture, …), and the hyperparameters to chosen (for example aggregators used for the selection and aggregation phases of Bulyan), the optimization method the attacker uses for designing an attack, and the implementation details (code efficiency).
>
> Empirically, as shown in Figure 3, designing a successful attack given the structure of the aggregation rule is computationally easier compared to designing an attack against a random pool of different aggregators with various structures. Here, we apply random resampling along all methods including comed, omniscient, and krum.
>
> We have elaborated on this in Section 4.1 of the uploaded revision, and would be happy to clarify if you have any further concerns.
>
> ------
>
> - Empirical evaluation only compares with 2 prior works?:
>
> In Figure 4, we compare MixTailor with 5 baselines under both random-$\epsilon$ attack and more number of Byzantine workers. If the reviewer has any specific suggestion on baseline to compare, we would be happy to consider them in our revised paper.
>
> ------

---

> ### Author Response · Authors · 2022-09-01
> **Response to Reviewer mqSq Part 2**
>
> - Compare with [Allen-Zhu et al., 2021, Data and Diggavi, 2021, Gorbunov et al., 2022, Zhu et al., 2022]?:
>
> In Section 1.2 of the revision, we cite all of them. Allen-Zhu et al. (2021) have proposed a method, which requires additional memory to keep the state of workers, i.e., their gradients and particular estimate sequences over the course of optimization. Such state vectors  are used to update the set of good workers at each iteration. In their method, the server has to keep track of the history of updates by each individual user, which is not practical in large-scale systems. Such additional memory is not required for MixTailor.
>
> Data and Diggavi, 2021 have proposed a robust method based on robust mean estimation in high dimensions and local SGD. Their method is built upon Robust Accumulated Gradient Estimation (Diakonikolas et al., 2019), which is computationally expensive and assumes correct samples are evenly distributed in all directions. As stated in Remark 1, optimizing over the number of local updates $\tau$ is a challenging problem and, to the best of our knowledge, there are very special problems for which local SGD is provably shown to outperform minibatch SGD. MixTailor is a plug and play scheme, which is compatible with local updating and fine tuning tricks to further improve communication efficiency and fairness.
>
> Please kindly note that both [Gorbunov et al., 2022] and [Zhu et al., 2022] are concurrent work. In an independent and concurrent work, Gorbunov et al., 2022 proposed a variance-reduced robust method which requires stringent assumptions including bounded global Hessian variance and local Hessian variance. Such assumptions are not required for MixTailor. Beyond assumptions,  standard variance-reduced methods are known to fail in terms of showing variance reduction for the majority of each epoch when applied to high-capacity neural networks, which makes them hard to train for practitioners [Defazio and Bottou, 2019]. It is beyond the scope of our paper but we will be happy to elaborate on this issue if the reviewer is interested.
>
> Finally, in an independent and concurrent work, Zhu et al., 2022 have improved the dimension dependence and achieved a tight statistical rate in terms of all the parameters for strongly convex losses. Though an important class of loss functions, the assumption of strongly convex loss functions does not hold in practice even for the case of shallow neural networks. MixTailor does not have such restrictions.
>
>
>
> [Allen-Zhu et al., 2021] Zeyuan Allen-Zhu, Faeze Ebrahimianghazani, Jerry Li, and Dan Alistarh. Byzantine-resilient non-convex stochastic gradient descent. In Proc. International Conference on Learning Representations (ICLR), 2021.
>
> [Data and Diggavi, 2021] Deepesh Data and Suhas Diggavi. Byzantine-resilient high-dimensional SGD with local iterations on heterogeneous data. In Proc. International Conference on Machine Learning (ICML), 2021.
>
> [Gorbunov et al., 2022] Eduard Gorbunov, Samuel Horvath, Peter Richtarik, and Gauthier Gidel. Variance reduction is an antidote to Byzantines: better rates, weaker assumptions and communication compression as a cherry on the top. arXiv preprint arXiv:2206.00529.
>
> [Zhu et al., 2022] Banghua Zhu,  Lun Wang, Qi Pang, Shuai Wang, Jiantao Jiao, Dawn Song, and Michael I. Jordan. Byzantine-robust federated learning with optimal statistical rates and privacy guarantees. arXiv preprint arXiv:2205.11765.
>
> [Defazio and Bottou, 2019] Aaron Defazio and Leon Bottou. On the ineffectiveness of variance reduced optimization for deep learning. In Advances in Neural Information Processing Systems (NeurIPS), 2019.

---

> ### Author Response · Authors · 2022-09-18
> **New Revision**
>
> Dear Reviewer mqSq,
>
> We just noticed that comparisons with [Allen-Zhu et al., 2021, Data and Diggavi, 2021, Gorbunov et al., 2022, Zhu et al., 2022] were missed in an earlier revision. We just submitted a new revision and made sure all those papers are cited in Section 1.2. We would be happy to clarify if you have any further concerns.
>
> Regards,
>
> Authors

---

### Review · Reviewer_dSxc · 2022-08-01

**Summary Of Contributions:**

The submission "MixTailor: Mixed Gradient Aggregation for Robust Learning Against Tailored Attacks" investigates byzantine-resistent distributed training of machine learning models. A range of robust aggregations strategies have been proposed in the literature so far, however these can be broken by specialized attackers. This submission instead proposes to randomly draw an aggregation strategy from a pool of known aggregation strategies to thwart potential attacks. The submission provides conditions under which this scheme still converges in i.i.d and non-i.i.d scenarios and provides experimental evidence that attacks proposed in Fang2020, Xie2020 and Baruch2019 cannot break the proposed randomized aggregation scheme.

**Broader Impact Concerns:**

No broader impact concerns from my side.

**Requested Changes:**

Requested changes:

My main concern is with the strength of the considered attack. The paper sets out claiming that other work creates defenses that only defend against known attacks, but then falls into the same trap and only considers existing attacks with minor modifications. This could be addressed in several ways.
* Additional theoretical results and exposition discussing why attacks breaking multiple aggregation rules are impossible.
* Empirical construction of vectors that fulfill all/most aggregation rules and showcasing of their limited effectiveness.
* Additional exposition and highlighting of the issue of optimal attacks and potential for an optimal attacker to break the proposed scheme in this way.


Otherwise I would welcome a rewriting of the exposition to fix the minor weaknesses listed above, but this is not as important of a change.

Further, there are several typos which I will list below:
* systems creates new vulnerabilities
* The model position attack is
* designed with a prior knowledge
* variants were well studied
* proposed secure aggregation
* and proposed resampling idea to
* we used resmpling before

**Strengths And Weaknesses:**



Overall this is a clever submission that is clearly written and nicely relates to work done in byzantine-robust distributed training in the last years. The additional results and generalizations of Krum are also helpful to the wider community. My main criticism with this submission as it stands related to the strength of the investigated attack and the assumptions related to it. I'll provide more details below:



A central tenet of this submission is that "MixTailor makes the design of successful attack strategies extremely difficult, if not impossible". However, in the current format of this manuscript this issue is not as clear-cut as it should be. If my reading of the experimental section is correct, the attacks evaluated against the proposed randomized aggregation rule are using baseline attack hyperparameters originally proposed in Fang2020/Xie2020, with attacks happening either at eps=0.1 or eps=10. It is not clear to me why this would be the optimal attack against the proposed aggregation rule.

Given that all aggregators are estimators of the gradient of the true empirical risk, it would follow that the intersection of updates satisfying all aggregators is non-empty - it contains at least the true gradient in expectation. So what is the performance of an attack that fulfills the constraints for multiple aggregations? The existing theoretical analysis in Prop.1 is very interesting, but I would have liked to see a closer look at the possibility or impossibility of a stronger attack, i.e. of attacks that violate the constraints imposed by the assumptions of Prop.1. I think the authors have thought about this, given Remark2, as well as the additional attack objectives mentioned in appendices A.1 and A.4 - albeit with limited descriptions -, as well as the reference to lower bounds in Xie vs upper bounds in Fang, but to me, this is really the crux of the article and should be given more space. For example, by restating these bounds and proving when their intersection is empty?
The best attack might only in expectation fulfill the constraint set spanned by all aggregation rules, but an attack that fulfills most constraints could already be much stronger than the baseline attack chosen in the experimental section. The fact that existing attacks against Krum can be defended against with comed is insufficient evidence about the strength of a joint attack.


I further see a few weaknesses in the exposition, which I will also briefly summarize below>:

* In the introduction and related work, the submission spends a good amount of space discussing backdoor attacks. However, everything that follows later on does not discuss backdoor attacks again. To clarify, let's assume the nomenclature of Biggio2012 which distinguishes between "availability" and "integrity" attacks. Backdoor attacks, using patches or edge-cases then fall under the category of integrity attacks, whereas attacks that reduce model performance in general fall under the category of availability atatcks. The tailored attacks discussed in this work are strictly availability attacks and reduce model performance. Conversely, the investigated robust aggregation schemes from byzantine learning are used to defend against model availability attacks. The tailored attack is clearly not an optimal backdoor attack, and it is in general unclear whether robust aggregation schemes would defend against backdoor attacks (or rather, depending on the definition of backdoor attack, this is undecidable). The current layout of the related work section gives, atleast me,  the misleading impression that this work would investigate backdoor attacks and defenses and that the proposed attack would defend against backdoor attacks. To be clear, I do not think this is necessary and good defenses against availability attacks are important, my problem is only with the weight of this section, and limited commentary.

* The tailored attack discussed in Sec.2.3 is ultimately a greedy attack, providing a direction towards maximal loss at the current iterate. In Fang2020, this was shown to be sufficient for an attack against several aggregation rules. Yet, overall this is not necessarily the optimal attack.

* In the related work section, it is mentioned that robust and secure aggregation are complementary. I would argue (and this argument is also brought up in the conclusions) that robust and secure aggregation are currently strictly at odds, given that secure aggregation is currently only feasible as mean-aggregation, which reduces the potential impact of a now much smaller robust aggregation of these means.

* The abstract mention "multi-GPU" systems as being under threat, but this point is never picked up again in the manuscript, and as it is, incorrect? Multi-GPU systems that are not distributed across machines do not appear to be affected by this attack.

* The variable f is reused for both individual functions f_i and number of compromised workers

* Otherwise the submission contains several typos (see below).

---

> ### Author Response · Authors · 2022-09-01
> **Response to Reviewer dSxc Part 1**
>
> We thank the reviewer for their thoughtful comments which we address one by one below.
>
> ------
>
> - Given that all aggregators are estimators of the gradient of the true empirical risk, … the intersection of updates satisfying all aggregators … contains at least the true gradient in expectation
>
> We note that median-based aggregators **do not necessarily output an unbiased estimate of the gradient of the true empirical risk** [Karimireddy et al., 2021, Section 3]. One simple example is to use median-based methods to estimate the proportion parameter $p$ of a Bernoulli distribution. For simplicity let $p=1/2$ and suppose we are given $N$ random samples with an odd $N$. The true mean is $1/2$.  For an $N$ odd, median-based method such as  Krum, comed, and Bulyan will always return either 0 or 1, which is a biased estimate of the true mean. The bias does not go to zero even if $N$ goes to infinity, all samples are honest, and we take expectation w.r.t. the original samples. In the revision, we have clarified this in Appendix A.7.
>
>
> ------
>
> - performance of an attack that fulfills the constraints for multiple aggregations
>
> Using both theoretical analysis and empirical results, we show that MixTailor increases complexity of attack design for an informed adversary:
>
> **Theoretical analysis**
>
> Suppose that the $M$ aggregators used by the server include $\{\mathrm{AGG}_1, \mathrm{AGG}_2, \ldots, \mathrm{AGG}_M\}.$
>
>
> We note that each aggregation rule $\mathrm{AGG}_i$ is either deterministic or has some hyperparameters which can be set randomly such as sub-sampling parameters. For each $\mathrm{AGG}_i$, we define attack complexity as follows:
>
> Let $T_i(n,f,d,\epsilon)$ denote the *number of elementary operations* an informed adversary requires to design a tailored attack in terms of solving the optimization problem in Section 2.3 with precision $\epsilon$ for satisfying constraints such that $\epsilon=\arg\cos\Big(\frac{-{\bf g}^{\top}{\bf g}’}{\|{\bf g}’\|\|{\bf g}\|}\Big)=\pi-\arg\cos\Big(\frac{{\bf g}^{\top}{\bf g}’}{\|{\bf g}’\|\|{\bf g}\|}\Big)$. For a given aggregation rule, the number of elementary operations increases to achieve smaller values of $\epsilon$, which amounts to optimizing more effective attacks. The attack complexity for MixTailor is ${\Omega}(\sum_{i=1}^M T_i\big(n,f,d,\epsilon)\big)$, which monotonically increases by $M$. To see this, assume there exists an attacker with lower complexity, then the attacker fails to break at least one of the aggregators. Note that precise expressions of $T_i$’s, i.e., the exact numbers of elementary operations depend on the specific problem to solve (dataset, loss, architecture, …), and the hyperparameters to chosen (for example aggregators used for the selection and aggregation phases of Bulyan), the optimization method the attacker uses for designing an attack, and the implementation details (code efficiency).
>
> We have elaborated on this in Section 4.1 of the uploaded revision, and would be happy to clarify if you have any further concerns.
>
> **Empirical results**
>
> We ran an additional experiment with eps=5.05 which is the average of 0.1 and 10 epsilons that were shown to be effective against Krum and Comed. The experiment is performed on the MNIST dataset in IID setting. There are 12 workers in total where 2 of them are Byzantine workers. The result is shown in the table below. This attack is not as effective as tailored attacks against any specific rule.
>
> | Aggregator  Accuracy |
> | -----  |  ----- | ------ |
> |Omniscient | 97.62-97.68% |
> |MixTailor|95.61-97.26% |
> |Krum| 98.34-98.46% |
> |Comed| 87-91% |
>
>
>
> ------

---

> > ### Comment · Reviewer_dSxc · 2022-09-07
> > **Updates**
> >
> > Thank you for providing these additional clarifications. A few comments,
> > * The comment noting that the median-based aggregators are not necessarily unbiased estimates is relevant, but my underlying question remains, whether it could be disproven that the "the intersection of updates satisfying all aggregators is non-empty ". I now agree that this does not follow from unbiasedness of the estimator, but the question remains.
> > * The theoretical analysis is great, and I agree that the proposed scheme will monotonically improve the attack complexity in terms of elementary operations. Yet, this is a much weaker claim than the overall message of the paper (for example, compared to the first sentence of the conclusions).
> > * I am unconvinced by the empirical results discussed in the response above. Running the attack of Fang2020 with eps scaled to 5.05 is not necessarily a sensible attack against both Krum and Comed. That the attack of Fang2020 cannot be easily adapted to be successful against these defenses is no evidence that no attack against both defenses exist.
> > * That being said, the remark concerning the intersections of both bounds is very helpful (Although it would be better if the derived condition could be exemplified for practical values from the experimental section.)
> > * Furthermore thank you for clarifying the exposition. I do wonder whether the whole block of related work concerning backdoor attack is still necessary. This topic is currently discussed and then immediately discarded, whereas availability attacks such as in Fang and Xie and relegated to a part of the related work for robust aggregation.

---

> > > ### Author Response · Authors · 2022-09-12
> > > **New Revision**
> > >
> > > We thank the reviewer for quickly engaging with us and providing helpful suggestions, which improve the quality of the paper and clarity of presentation.
> > >
> > > ------
> > >
> > > -  Intersection of updates satisfying all aggregators is non-empty
> > >
> > > We agree. Depending on the set of aggregators, the intersection of sets of tailored attacks against those aggregators may be non-empty. As the size of the set of underlying aggregators in MixTailor grows, the intersection set shrinks, and the computational complexity of finding such solutions increases monotonically.
> > >
> > > ------
> > >
> > > - Revise the claim in the first sentence of the conclusions
> > >
> > > We have revised the first sentence of conclusions and the third paragraph of Section 3 to match our theoretical analyses and clarified that "To increase computational complexity of designing tailored attacks for an informed adversary, we introduce MixTailor based on randomization of robust aggregation strategies."
> > >
> > > ------
> > >
> > > - The attack of Fang2020 with $\epsilon$ scaled to 5.05 is not necessarily a sensible attack against both Krum and Comed
> > >
> > > As shown in Figure 1c, when $\epsilon=0.1$ the attacker is able to corrupt both Krum and Comed on CIFAR-10. For a general setting, we construct an attack that minimizes dot product with the true gradient w.r.t. a number of aggregators. In Section 5 of the revision, we have considered a stronger and adaptive attacker, which optimizes its attack by enumerating over a set of $\epsilon$'s and selects *an adaptive $\epsilon$ against the aggregator at every single iteration*. The adversary can apply such adaptive attack at the expense of significantly higher computational costs.
> > >
> > > ------
> > >
> > > - Is block of related work regarding backdoor attacks necessary?
> > >
> > > In the revision, we have removed discussion on backdoor attacks and provided a brief paragraph regarding "Data poisoning and model poisoning".
> > >
> > > ------

---

> > > > ### Comment · Reviewer_dSxc · 2022-09-13
> > > > **New Revision**
> > > >
> > > > Thank you for the update and additional clarifications!

---

> ### Author Response · Authors · 2022-09-01
> **Response to Reviewer dSxc Part 2**
>
> - possibility or impossibility of attacks that violate … Assumptions of Prop.1
>
> Let $\hat{\bf w}\in\mathbb{R}^d$.  To fail the conditions specified in Prop. 1, an adversary should have sufficient computational resources to find an attack (if exists) such that $\mathbb{E}[U_i(\hat{\bf w})]^\top\nabla F(\hat{\bf w})<0$ for $i\in[q]$ where *$q$ should be large enough*. Suppose that the adversary has sufficient random samples from each honest client to compute the expectation over the output of an aggregation rule $\mathrm{AGG}_i$ and has access to an accurate estimate of $\nabla F(\hat{\bf w})$. An aggregation $\mathrm{AGG}_i$ is typically a *nonconvex function* of the attack ${\cal B}$ in Prop. 1. Instead of designing an optimal attack, suppose that the adversary plans to verify an attack, which is a computationally simpler problem. By verification, we mean computing the output of $\mathrm{AGG}_i$ under an attack ${\cal B}$ and computing the sign of $\mathbb{E}[U_i(\hat{\bf w})]^\top\nabla F(\hat{\bf w})$. The verification runtime increases monotonically as $q$ increases. We note that due to **nonconvexity of baseline aggregation rules** such as comed, Krum, ..., we are unaware of any polynomial time algorithm with provable guarantees to efficiently corrupt multiple aggregation rules at the same time. With a similar argument as Section 4.1 (attack complexity) of the uploaded revision, MixTailor increases the complexity of attack design for an informed adversary.
>
> In the revision uploaded, we have clarified this in Remark 3.
>
> ------
>
> - Restate lower bounds in [Xie et al., 2020b, Theorem 1] and upper bounds in [Fang et al., 2020, Theorem 1] and prove when their intersection is empty?
>
> Thanks for this insightful comment. In [Fang et al., 2020, Theorem 1],  an upper bound is established on the norm of the attack vector that is tailored against Krum. In particular, $\lambda={\cal O} (1/\sqrt{d})$ fails Krum. Let $\tilde{\bf g}=\frac{1}{n-f}\sum_{i=f+1}^n{\mathbb{E}[\bf g_i]}$ denote expected value of honest updates sent by good workers. Building on a similar argument as in [Xie et al., 2020b, Theorem 1] and [Hawkins, 1971, Theorem 1(b)], a lower bound on $\lambda$ tailored against comed is given by $\lambda=\Omega\Big(\Big|1-\frac{\hat\sigma}{\sqrt{n-f-1}\|\tilde{\bf g}\|_{\infty}}\Big|\Big)$ where $\hat\sigma$ is the coordinate-wise variance defined in [Xie et al., 2020b, Theorem 1].
>
> A sufficient condition that guarantees emptiness of the intersection for an attack, which fails both Krum and comed is that the *variance $\hat\sigma$ is large enough* such that $\frac{\hat\sigma}{\sqrt{n-f-1}\|\tilde{\bf g}\|_{\infty}}\geq 1-\frac{1}{\sqrt{d}}$.
>
> In the revision uploaded, we have clarified this in Appendix A.4. We hope this answers your question and we would be happy to clarify if you have any further concerns.
>
>
> [Hawkins, 1971] Douglas M. Hawkins. On the bounds of the range of order statistics. Journal of the American Statistical Association, 66:644–645, 1971.
>
> ------
>
> - Clarify that this work focuses on availability attacks and does not investigate backdoor attacks.
>
> Thanks for clearly pointing out the formal distinction between “availability” and “integrity” attack. In Section 1.2 (related work) of the revision uploaded, we have clarified that “In this paper, we focus on tailored training-time attacks, which belong to the class of poisoning availability attacks based on the definition of Demontis et al., (2019). We do not study poisoning integrity attacks [Demontis et al., 2019], and MixTailor is not designed to defend against backdoor or edge-case attacks aiming to modify predictions on a few targeted points.”
>
> [Demontis et al., 2019] Ambra Demontis, Marco Melis, Maura Pintor, Matthew Jagielski, Battista Biggio, Alina Oprea, Cristina Nita-Rotaru, and Fabio Roli. Why do adversarial attacks transfer? explaining transferability of evasion and poisoning attacks. In Proc. USENIX security symposium (USENIX security 19), 2019.
>
> ------
>
> - The tailored attack in Sec.2.3 is ultimately a greedy attack, not necessarily the optimal attack.
>
> In Section 2.3 of the revision, we clarify that “This tailored attack is shown to be sufficient against several aggregation rules [Fang et al. (2020)]; however, it is not necessarily an optimal attack.”
>
> ------
>
> - Secure aggregation is currently only feasible as mean-aggregation.
>
> In the revision uploaded, we removed “Robust and secure aggregations can be viewed as complementary technologies for practitioners …” from the related work.
>
> ------
>
> - Multi-GPU systems that are not distributed across machines do not appear to be affected by this attack.
>
> We agree. In the abstract of the revision uploaded, we removed “Multi-GPU systems” to avoid any confusion.
>
> ------

---

> ### Author Response · Authors · 2022-09-01
> **Response to Reviewer dSxc Part 3**
>
>
> - Variable $f$ is reused for both individual functions $f_i$ and number of compromised workers.
>
> Thanks for pointing out this issue. In the revision, we use capital $F_i$ in Eq. (1) to improve presentation of the paper.
>
> ------
>
> - Fix the typos.
>
> Thanks for carefully reading our paper and helping us identify these typos. We have fixed them in the revision.
>
> ------

---

### Review · Reviewer_Gy9m · 2022-08-10

**Summary Of Contributions:**

**[Summary of the Paper]**

This paper presents a defense mechanism against the attacker who wants to cause the accuracy loss in the distributed learning settings by controlling Byzantine workers. The paper’s insight is that, as existing countermeasures, such as Krum, use statistical differences, e.g., a large variance constructed from SGD computations, they have been circumvented by advanced attacks. Hence, the paper proposes a technique that uses an aggregation rule drawn from a set of robust aggregation rules uniformly at random, increasing the difficulty of the attacker knowing the actual aggregation rule used at the i-th round. In evaluation with MNIST and CIFAR-10 benchmarks, the paper demonstrates its effectiveness at mitigating the impact of Byzantine gradients generated by the adversary, compared to the four baseline mechanisms, i.e., comed, Krum, and two others proposed by Pillutla et al., and El Mhamdi et al. in both i.i.d and non-i.i.d settings.

**[Contributions]**

1. Presents an aggregation rule against Byzantine attackers in the distributed settings, more robust to the baselines.
2. Shows its effectiveness in MNIST and CIFAR-10 benchmarks, in both i.i.d and non-i.i.d settings.

**Broader Impact Concerns:**

None.

**Requested Changes:**

1. Consider adaptive attackers
- 1.1 vs. who knows all the defenses in the set M and wants to evade them all at once.
- 1.2 vs. who is dumb and just do the same attacks that evades one of defenses in M until it's successful
- 1.3 vs. who uses a few rounds as a steppingstone and makes the accuracy drop at a certain point
- 1.4 vs. more adversary who wants to break this ensembling-like defense.

2. Revisit theoretical guarantees
- 2.1 formally evaluates the computational costs for that convergence guarantee
- 2.2 formally and empirically evaluates the increase of computational costs as we consider more and more aggregation mechanisms.
- 2.3 (formally if possible) and empirically evaluates whether the computational increase will be bounded

3. Revisit evaluation
- 3.1 Remove backdoor attack section if it's not relevant to the paper
- 3.2 Tone down the claim and clarify that the defense can defeat an adversary who aims to degrade the accuracy.
- 3.3 Show the evaluation that I suggested in 2.


**Strengths And Weaknesses:**

**[Strengths]**

1. Well-written and easy-to-follow.
2. Proposes a simple, yet effective countermeasure against the informed attacker.

**[Weaknesses]**

1. Not considering a sufficiently strong adversary.
2. Missing considerations for the computational cost for the guarantee.
3. Not sure evaluations are sufficient for supporting contributions.

**[Detailed Comments]**

This is an interesting paper asking how we can make the aggregation mechanism more robust against advanced attackers. I am a bit reluctant to see that it is possible, but understanding the limit is worth exploring. However, regarding the completeness of this study, I have several concerns that should be addressed before acceptance. I listed my concerns in detail below:

*Consideration of an adaptive attacker*

(1) If I am an adversary who wants to compromise the training, I will construct gradient updates that can evade a set of aggregation mechanisms (M) that the defender may use. Once the attacker finds the gradient updates that can degrade the utility of the central model while evading M defenses, pulling a defense uniformly at random at each round does not prevent the attacker from compromising the final model.

(2) In addition to that, I am a bit concerned that even a dumb attacker can evade this defense. Suppose that I consistently attack the model with a mechanism M'. If the defender considers five mechanisms one out of five times, my attack can successfully degrade the model's accuracy. If the attacker causes an accuracy drop of 50% at round 7, how long do we have to run more rounds to recover the accuracy of the dropped model? If it's *not* smaller than 5 (the number of mechanisms this defense considers), I think the attacker always has the upper hand.

(3) Moreover, this attacker does not have to be successful within five rounds (in contrast to the paper's experiments). As an adversary, I would inject the gradient updates that degrade the accuracy over multiple rounds (e.g., 0.5% in each round) or at a specific round (e.g., the model is fine until round 50 and, at the 51st, the model degrade the accuracy by 10%).

Without all those adaptive attacks, I am a bit concerned that the defense couldn't be better than just deploying one existing defense.


*Theoretical Guarantees*

We can expect that this defense mechanism anyway converges. A vast literature on the attacks and defenses against deep neural networks shows that a model's training can achieve a high accuracy even when 10% of the training data is compromised, i.e., the labels are corrupted. Hence, it may be a bit obvious to talk about the convergence here.

(1) What I am more curious about is that convergence comes at the same computational costs as baseline defenses. As an example, if this defense takes 5x times to converge compared to the case where we use Krum, it is hard to say the defense is practical.

(2) We can further consider the interaction between the number of aggregations that this defense considers vs. the time it takes to converge to reasonably high accuracy. Suppose that we have to consider 100 defenses. I wouldn't expect the computational costs to increase to 100x times.


*Weak Evaluation*

I am a bit worried that this paper is not clear about the adversary's objective throughout the paper. The paper considers a specific adversary who wants to degrade the accuracy of the final model. However, some sections in the paper offer an impression that the paper can defeat most adversaries in distributed learning settings.

In particular, I am unsure about the use of "backdoor attacks" in Section 1.2. It's a very different adversary who compromises both the training and testing samples for successful attacks. For example, unless this paper evaluates the attacks that cause misclassification of specific samples or backdooring, and so on, I do believe that the point should be clear.

In addition to that, as stated in the previous paragraphs, the paper must evaluate the defense against adaptive adversaries and the computational costs theoretically and empirically, when comparing to the baselines.

---

> ### Author Response · Authors · 2022-09-01
> **Response to Reviewer Gy9m Part 1**
>
> We thank the reviewer for their thoughtful comments which we address one by one below.
>
> ------
>
> - Not considering a sufficiently strong adversary. Consider an adaptive attacker
>
> In the revision, we have considered a stronger and adaptive attacker which optimizes its attack by enumerating over a set of $\epsilon$’s and selects **the worst $\epsilon$ against the aggregator at every single iteration**. The adversary enumerates among all those $\epsilon$’s and finds out which one is the most effective attack by applying the aggregator (the attacker simulates the server job by applying the aggregator with different $\epsilon$’s and finds the best attack and then outputs the best attack for the server to aggregate). Regarding MixTailor, the attacker selects a random aggregator from the MixTailor's aggregator pool in each iteration and finds the worst epsilon corresponding to this aggregator. The attacker finds the worst attack by calculating the dot product of the output of the aggregator and the direction of aggregated gradients without attacks when different epsilons are fed into the aggregator. The attacker chooses the epsilon that causes the aggregator to produce the gradient that has the smallest dot product with the true gradient. We ran this experiment over the MNIST dataset, and observed that MixTailor is able to outperform both Krum and Comed. Comed's accuracy changes between 85-87%, Krum's accuracy is 83-85%, and MixTailor's accuracy is 91.80-92.55%. The accuracy of the omniscient aggregator is 97.62-97.68%. The set of epsilons used by the adaptive attacker is 0.1, 0.5, 1, and 10.
>
> ------
>
> - Consider an attacker that evades $M$ defenses
>
> Let $\hat{\bf w}\in\mathbb{R}^d$. To fail multiple defense aggregators and fail the conditions specified in Proposition 1, an adversary should have sufficient computational resources to find an attack (if exists) such that $\mathbb{E}[U_i(\hat{\bf w})]^\top\nabla F(\hat{\bf w})<0$ for $i\in[q]$ where *$q$ should be large enough*. Suppose that the adversary has sufficient random samples from each honest client to compute the expectation over the output of an aggregation rule $\mathrm{AGG}_i$ and has access to an accurate estimate of $\nabla F(\hat{\bf w})$. An aggregation $\mathrm{AGG}_i$ is typically a *nonconvex function* of the attack ${\cal B}$ in Proposition 1. Instead of designing an optimal attack, suppose that the adversary plans to verify an attack, which is a computationally simpler problem. By verification, we refer to computing the output of $\mathrm{AGG}_i$ under an attack ${\cal B}$ and computing the sign of $\mathbb{E}[U_i(\hat{\bf w})]^\top\nabla F(\hat{\bf w})$. The verification runtime increases monotonically as $q$ increases. We note that due to **nonconvexity of baseline aggregation rules** such as comed, Krum, ..., we are unaware of any polynomial time algorithm with provable guarantees to efficiently corrupt multiple aggregation rules at the same time.
> In the revision uploaded, we have clarified this in Remark 3.
> We now show that MixTailor structure increases complexity of attack design for an informed adversary: Suppose that the $M$ aggregators used by the server include $\{\mathrm{AGG}_1, \mathrm{AGG}_2, \ldots, \mathrm{AGG}_M\}.$
>
>
> We note that each aggregation rule $\mathrm{AGG}_i$ is either deterministic or has some hyperparameters which can be set randomly such as sub-sampling parameters. For each $\mathrm{AGG}_i$, we define attack complexity as follows:
>
> Let $T_i(n,f,d,\epsilon)$ denote the *number of elementary operations* an informed adversary requires to design a tailored attack in terms of solving the optimization problem in Section 2.3 with precision $\epsilon$ for satisfying constraints such that $\epsilon=\arg\cos\Big(\frac{-{\bf g}^{\top}{\bf g}’}{\|{\bf g}’\|\|{\bf g}\|}\Big)=\pi-\arg\cos\Big(\frac{{\bf g}^{\top}{\bf g}’}{\|{\bf g}’\|\|{\bf g}\|}\Big)$. For a given aggregation rule, the number of elementary operations increases to achieve smaller values of $\epsilon$, which amounts to optimizing more effective attacks. The attack complexity for MixTailor is ${\Omega}(\sum_{i=1}^M T_i\big(n,f,d,\epsilon)\big)$, which monotonically increases by $M$. To see this, assume there exists an attacker with lower complexity, then the attacker fails to break at least one of the aggregators. Note that precise expressions of $T_i$’s, i.e., the exact numbers of elementary operations depend on the specific problem to solve (dataset, loss, architecture, …), and the hyperparameters to chosen (for example aggregators used for the selection and aggregation phases of Bulyan), the optimization method the attacker uses for designing an attack, and the implementation details (code efficiency).
>
> We have elaborated on this in Section 4.1 of the uploaded revision, and would be happy to clarify if you have any further concerns.
>
> ------

---

> ### Author Response · Authors · 2022-09-01
> **Response to Reviewer Gy9m Part 2**
>
> - Consider an attacker who uses a few rounds as a steppingstone and makes the accuracy drop at a certain point. If the attacker causes an accuracy drop of 50% at round 7, how long … more rounds to recover …?
>
> We theoretically and empirically show that such an attack is not stronger than the attacks already considered in the paper. Theoretically, if the attacker injects its poison at a particular iteration $t$, for example at round 7, the  **maximum parameters’ deviation** is upper bounded $L\eta_t$ where $\eta_t$ is the step size in Section 2.2 and $L$ is the maximum gradient norm of the loss function $\ell(\cdot,{\bf z})$ in (1).  The **maximum loss increase** after an attack is upper bounded by $L^2\eta_t$.  Note that $\eta_t$ is typically very small when training for example neural networks, which decays over the course of training. In practice, for example ResNets are observed to have stable gradients throughout training [He et al., 2016, Hanin 2018, Zaeemzadeh et al., 2016]. In addition, a number of techniques have been proposed to control gradient norms including proper initialization schemes, proper activation functions, batch normalization, regularization, and gradient clipping [Pascanu et al., 2013, Klambauer et al., 2017, Hanin 2018, Zaeemzadeh et al., 2021].    Assuming $L$ is sufficiently small, the effective poison cannot change the loss and accuracy much at a single iteration. For a training-time attack to be effective, it should be applied consistently for some consecutive iterations. We empirically observe that MixTailor performance never drops by 50% regardless of the attack even when the attacker enumerates from multiple $\epsilon$ reverse attacks and tries to optimize over $\epsilon$ as shown in Figure 7.
>
> [He et al., 2016] Kaiming He, Xiangyu Zhang, Shaoqing Ren, and Jian Sun. Deep residual learning for image recognition. In Proc. IEEE Conference on Computer Vision and Pattern Recognition (CVPR), 2016.
>
> [Hanin 2018]  Boris Hanin. Which neural net architectures give rise to exploding and vanishing gradients? In Advances in Neural Information Processing Systems (NeurIPS), 2018.
>
> [Zaeemzadeh et al., 2021] Alireza Zaeemzadeh, Nazanin Rahnavard and Mubarak Shah. Norm-Preservation: Why Residual Networks Can Become Extremely Deep? IEEE Transactions on Pattern Analysis and Machine Intelligence (TPAMI), 43(11): pp. 3980-3990,  2021.
>
> [Klambauer et al., 2017] Gunter Klambauer, Thomas Unterthiner, Andreas Mayr, and Sepp Hochreiter. Self-normalizing neural networks. In Advances in Neural Information Processing Systems (NeurIPS), 2017.
>
> [Pascanu et al., 2013] Razvan Pascanu, Tomas Mikolov, and Yoshua Bengio. On the difficulty of training recurrent neural networks. In International Conference on Machine Learning (ICML), 2013.
>
>
> ------
>
> - Consider a naive adversary who evades one of defenses in $[M]$ until it's successful
>
> We have already observed that MixTailor is robust to an attack tailored against one of the aggregators, for example in Figures 1, 2, 3, and 4 in the original paper.
>
> ------
>
> -  How does increasing the number of adversaries enhance an attack's performance?
>
>
> Theoretically, for a fixed $M$ number of candidate aggregators in the pool, in Proposition 1, we established an upper bound on the number of failed rules $q$ and showed that MixTailor is robust as long as $q$ is not very large. In the revision, we have elaborated on this in Remark 3.
>
> Empirically, Fig. 4b shows the results with more Byzantines and shows that geomed may be vulnerable to such attacks designed for Krum and comed. MixTailor always outperforms the worst aggregator, which is the target of a tailored attack.
>
>
>
> ------

---

> ### Author Response · Authors · 2022-09-01
> **Response to Reviewer Gy9m Part 3**
>
> - Convergence is not surprising. Deep neural networks .. achieve a high accuracy even when 10% of the training data is compromised, i.e., the labels are corrupted
>
> Deep neural networks are great feature extractors that can memorize training datasets. In particular, first-order methods with random initialization can consistently achieve zero training error, even with randomized labels [Zhang et al., 2017]. Such an empirical success can possibly be explained by the overparameterization of neural networks.
>
> With convergence, we refer to convergence to an empirical risk minimizer of the original objective (1) of honest workers. Please note that in this paper, we focus on model poisoning (training-time attack), not data poisoning. Our goal is to show that MixTailor is robust to attacks that are shown to fail well-known aggregation rules such as Krum and Comed. Our paper showed that compromised workers return an arbitrary vector such that Krum and Comed **converge to an ineffective model even if it converges.** Examples are shown in Figures 1, 2, 3, and 4 in the original paper. In Proposition 1 and Theorem 3, we find conditions, for example an upper bound on the number of failed aggregation rules under an attack, under which MixTailor is guaranteed to **converge to  an empirical risk minimizer of the original objective of honest workers**, i.e., converge to an effective model, which is not trivial. In the revision, we have clarified this after Theorem 3.
>
> [Zhang et al., 2017] Chiyuan Zhang, Samy Bengio, Moritz Hardt, Benjamin Recht, and Oriol Vinyals. Understanding deep learning requires rethinking generalization. In Proc. International Conference on Learning Representations (ICLR), 2017.
>
>
> ------
>
>
> - Evaluate the computational costs and the increased costs as we consider more and more aggregation mechanisms
>
> The analysis of computational complexity of MixTailor is discussed in Appendix A.2. In particular, the worst-case computational cost of MixTailor is upper bounded by that of the candidate aggregation rule with the maximum number of elementary operations per iteration. The average computation cost across the course of training is the average of the costs for candidate rules. The number of elementary operations per iteration for Bulyan with Krum as its aggregation rule is in the order of $O(n^2 d)$ [El Mhamdi et al., 2018]. The computational cost for coordinate-wise median and an efficient implementation of an approximate geometric median based on Weiszfeld algorithm is $O(n d)$ [Pillutla et al., 2022]. Increasing the number of aggregators in the pool does not necessarily increase the computation costs of MixTailor as long as the number of elementary operations per iteration for new aggregators is in the order of those rules that have been already in the pool of aggregators.
>
> We empirically provide computational costs for different aggregation rules after running 10 iterations. The following table shows the accuracy and Time per iteration for each aggregator used.
>
> | Aggregator  | Time per iteration (us) | Accuracy |
> | -----  |  ----- | ------ |
> |Omniscient | 60 | 97.62-97.68% |
> |MixTailor| 4980 |91.80-92.55% |
> |Krum| 2176 | 83-85% |
> |Comed| 153 | 85-87% |
>
> ------
>
> - Revise “backdoor attacks” in Section 1.2. and clarify that the defense can defeat an adversary who aims to degrade the accuracy
>
> In Section 1.2 (related work) of the revision uploaded, we have clarified that “In this paper, we focus on tailored training-time attacks, which belong to the class of poisoning availability attacks based on the definition of Demontis et al., (2019). We do not study poisoning integrity attacks [Demontis et al., 2019], and MixTailor is not designed to defend against backdoor or edge-case attacks aiming to modify predictions on a few targeted points.”
>
> [Demontis et al., 2019] Ambra Demontis, Marco Melis, Maura Pintor, Matthew Jagielski, Battista Biggio, Alina Oprea, Cristina Nita-Rotaru, and Fabio Roli. Why do adversarial attacks transfer? explaining transferability of evasion and poisoning attacks. In Proc. USENIX security symposium (USENIX security 19), 2019.
>
> ------

---

> > ### Comment · Reviewer_Gy9m · 2022-09-08
> > **Thank You for the Response**
> >
> > Thank the authors for the point-by-point response.
> > The authors' response mostly addresses my concerns.
> >
> > My primary concerns were:
> >
> > (1) As this is a simple yet seemingly effective defense, I'd like to see a discussion about adaptive attacks in the paper. It doesn't need to be the strongest, but it has to be sufficiently strong than the conventional attacks. I like their discussion of an adaptive attack in the last section of the revised paper.
> >
> > (2) Once we assume there's an adaptive attacker, I think it would be nice to emphasize some other advantages this defense can offer. Hence, I liked analyzing the computational cost that an adversary should spend to break the defense.
> >
> > (3) But the defense may also significantly increase the computational demands of an aggregation process; thus, I want potential readers to see the trade-offs more closely. I thank the authors for being honest about the time per iteration. It would have been better to see M as an x-axis and the time per iteration on the y-axis. But, I think the authors can add some writing to describe the increase (log-linear, linear, or exponential)
> >
> >
> > [Suggestions]
> >
> > I would tone down the claim "The attacker finds the worst attack by calculating the dot…" as we can define the worst-case multiple ways, e.g., retaining the performance degradations while reducing computational cost for achieving that.
> >
> > I wouldn't claim that "assuming L is sufficiently small, the effective poison cannot change the loss and accuracy much at a single iteration." Recent work showed that a small amount of parameter perturbations caused by 8-bit quantization, which is much smaller than the gradient updates this paper considers, can increase the performance degradation of a compressed model up to 99%. So, the claim cannot work for all the models. I also suggest the authors tone down a bit.

---

> > > ### Author Response · Authors · 2022-09-12
> > > **New Revision**
> > >
> > > We thank the reviewer for quickly engaging with us and providing helpful suggestions, which improve the quality of the paper and clarity of presentation.
> > >
> > > ------
> > >
> > > - Discuss adaptive attacks in the paper
> > >
> > > In Section 5 of the revision, we have provided a detailed discussion of the adaptive attack.
> > >
> > > ------
> > >
> > > - Clarify additional computational demands (log-linear, linear, or exponential)
> > >
> > >
> > > We have elaborated on this in Section 5 of the uploaded revision and clarified that "The average computation cost of MixTailor across the course of training is the average of the costs for candidate rules. As $M$ increases, the average time per iteration for MixTailor increases linearly with the average computation costs of $M$ underlying aggregators."
> > >
> > > ------
> > >
> > >
> > > - Tone down the claim "The attacker finds the worst attack …"
> > >
> > > In the revision, we have changed  it to "The attacker finds *an adaptive* attack by ... "
> > >
> > > ------
> > >
> > > - Recent work showed … small amount of parameter perturbations … can increase the performance degradation … up to 99%
> > >
> > >
> > > We agree. In the revision, we have clarified this in Section 6: "In this paper, we focus on settings where the maximum gradient norm of the loss across the course of training is sufficiently small such that the effective poison cannot change the accuracy much at a single iteration. Developing defense mechanisms in more challenging settings where an adversary is able to design an effective poison in one iteration is an interesting problem for future work."

---

### Comment · Action_Editors · 2022-08-27
**Authors?**

The authors are advised that it has been 17 days since all reviews have been submitted, and there is not yet any response by the authors or revisions based on the reviewer comments. Reviewers have already started to submit their official recommendations, and are required to submit their recommendations within the next 10 days. The authors are strongly recommended to engage with the reviewers, who have dedicated significant time to write very thoughtful reviews.

---

> ### Author Response · Authors · 2022-08-28
> **Revision**
>
> Dear Action Editor,
>
> Thanks for your note and handling our paper. We are finalizing our revision and plan to respond to all reviewers shortly. There are additional experiments requested by reviewers. We hope to receive the results in a couple of days, which complete our revision.
>
> Regards,
>
> Authors

---

### Comment · Action_Editors · 2022-09-30
**Camera Ready**

Hi Authors, thanks for submitting a camera ready version. In terms of content it should be fine, but I think there's a few style things that need to be changed? In particular, your header still says "Under review as submission to TMLR" and the "Reviewed on OpenReview:" is a placeholder. Can you please fix both of those? I'm not sure if these instructions were in the email you received.

---

> ### Author Response · Authors · 2022-10-01
> **Camera Ready Style**
>
> Dear Action Editor,
>
> Thanks for your quick message regarding the changes required for the style. We updated the camera ready and fixed the header. In the new version, it is shown "Published in Transactions on Machine Learning Research (10/2022)" and "Reviewed on OpenReview: https://openreview.net/forum?id=tqDhrbKJLS".
>
> Regards,
>
> Authors

---

### Decision · Action_Editors · 2022-09-20

**Recommendation:** Accept with minor revision

**Comment:**

The reviewers initially had some hesitations regarding the paper, but through the discussions with and revisions from the authors, were swayed positively. Specifically, the authors were skeptical about robustness of the method -- referring back to the TMLR acceptance criteria, this would have been a "no" answer the question "Are the claims made in the submission supported by accurate, convincing and clear evidence?". However, as some explorations in this direction, as well as modifications in the language, were performed by the authors, reviewers were satisfied and considered the paper above the bar for publication.

Reviewer Gy9m thought that reviewer dSxc had some good comments at the end, which it would be best if the authors could address. I notice that, after this, the authors revised the manuscript. I have marked this as "Accept with minor revision," but reviewer Gy9m can say whether they still feel additional revisions are necessary.